# Discrete populations of isotype-switched memory B lymphocytes are maintained in murine spleen and bone marrow

René Riedel [1,2], Richard Addo [1,13], Marta Ferreira-Gomes [1,13], Gitta Anne Heinz[1,13], Frederik Heinrich[1], Jannis Kummer[1], Victor Greiff[3,7], Daniel Schulz[1], Cora Klaeden[1], Rebecca Cornelis[1], Ulrike Menzel[3], Stefan Kröger[4,8], Ulrik Stervbo [1,5], Ralf Köhler[1], Claudia Haftmann[1,9], Silvia Kühnel[1], Katrin Lehmann[1], Patrick Maschmeyer[1], Mairi McGrath[1], Sandra Naundorf[1], Stefanie Hahne[1], Özen Sercan-Alp[1,10], Francesco Siracusa[1,11], Jonathan Stefanowski [1,5], Melanie Weber[1], Kerstin Westendorf[1], Jakob Zimmermann[1,12], Anja E. Hauser [1,5], Sai T. Reddy[3], Pawel Durek[1], Hyun-Dong Chang[1,13], Mir-Farzin Mashreghi [1,6,13✉] & Andreas Radbruch[1,13✉]

At present, it is not clear how memory B lymphocytes are maintained over time, and whether only as circulating cells or also residing in particular tissues. Here we describe distinct populations of isotype-switched memory B lymphocytes (Bsm) of murine spleen and bone marrow, identified according to individual transcriptional signature and B cell receptor repertoire. A population of marginal zone-like cells is located exclusively in the spleen, while a population of quiescent Bsm is found only in the bone marrow. Three further resident populations, present in spleen and bone marrow, represent transitional and follicular B cells and B1 cells, respectively. A population representing 10-20% of spleen and bone marrow memory B cells is the only one qualifying as circulating. In the bone marrow, all cells individually dock onto VCAM1$^+$ stromal cells and, reminiscent of resident memory T and plasma cells, are void of activation, proliferation and mobility.

[1] Deutsches Rheuma-Forschungszentrum (DRFZ), an Institute of the Leibniz Association, 10117 Berlin, Germany. [2] Evolutionary Genomics Group, Max Planck Institute for Evolutionary Biology, 24306 Plön, Germany. [3] Department of Biosystems Science and Engineering, Eidgenössische Technische Hochschule (ETH Zürich), CH-4058 Basel, Switzerland. [4] Knowledge Management in Bioinformatics, Humboldt-Universität zu Berlin, 12489 Berlin, Germany. [5] Charité-Universitätsmedizin Berlin, 10117 Berlin, Germany. [6] BCRT/DRFZ Single-Cell Laboratory for Advanced Cellular Therapies - Brandenburg Center for Regenerative Therapies (BCRT), 13353 Berlin, Germany. [7] Present address: Department of Immunology, Institute of Clinical Medicine, University of Oslo, 0424 Oslo, Norway. [8] Present address: Department of Infectious Disease Epidemiology, Robert Koch Institute, 13353 Berlin, Germany. [9] Present address: Institute of Experimental Immunology, Universitätsspital Zürich, 8057 Zürich, Switzerland. [10] Present address: R&D, TA Immunology & Inflammation Research, Sanofi-Aventis Deutschland GmbH, Industriepark Hoechst, 65926 Frankfurt am Main, Germany. [11] Present address: Department of General, Visceral and Thoracic Surgery, University Medical Center Hamburg-Eppendorf, 20246 Hamburg, Germany. [12] Present address: Department for BioMedical Research (DBMR), University of Bern, 3008 Bern, Switzerland. [13] These authors contributed equally: Richard Addo, Marta Ferreira-Gomes, Gitta Anne Heinz, Hyun-Dong Chang, Mir-Farzin Mashreghi, Andreas Radbruch. ✉email: mashreghi@drfz.de; radbruch@drfz.de

Memory B lymphocytes, together with memory T lymphocytes, embody one of the key features of the vertebrate immune system, the ability to memorize antigenic challenges once encountered. A comprehensive understanding of how memory B lymphocytes are maintained over time is key to understand the organization of immunological memory as such. Here we address the heterogeneity and localization of murine memory B lymphocytes. At present, it is an open question whether, and if so which populations of memory B cells are maintained as circulating cells and/or as resident cells of particular tissues, as has been described for tissue-resident memory T cells[1] and resident memory plasma cells[2,3].

Memory B cells expressing antibodies of switched isotype (Bsm) have been identified in different tissues and in the blood, and have been postulated to be part of one uniform, circulating population[4]. In humans, the spleen has been identified as a major hub for circulating *Vaccinia*-specific memory B cells[5] and splenectomy leads to gradual reduction in numbers of Bsm in the blood to about 50%[6], suggesting that tissues other than the spleen can also serve as hubs for circulating memory B cells.

Here we dissect the heterogeneity of Bsm of spleen and bone marrow (BM) of individual mice on the level of individual cells, and identify one circulating and several tissue-resident populations, the latter comprising the vast majority of Bsm in those organs. Overall, we find that most Bsm are located in the bone marrow and spleen of inbred and feral mice. Bsm of the BM and spleen of individual mice have significantly different repertoires, demonstrating that they constitute separate compartments. In the BM, Bsm rest in terms of proliferation and individually dock onto VCAM1$^+$/fibronectin$^+$ stromal cells, as described for memory T and memory plasma cells. Based on their transcriptomes, Bsm of BM can be separated into five different clusters, four of which are also found in the spleen. One cluster is exclusive to BM and represents IgG1$^+$ Bsm expressing little *Cr2* and *Fcer2a*, encoding CD21/35 and CD23, respectively. These Bsm are quiescent in terms of proliferation, transcription and activation, and their B cell receptors show an accumulation of somatic hyper-mutations. Very few if any of these Bsm are found in the spleen. Instead, in spleen an exclusive cluster of Bsm expressing IgG1 or IgG2, *Cr2* and *Fcer2a* is present, with transcriptomes resembling those of marginal zone B cells. Of the four Bsm clusters found in both spleen and BM, two have organ-exclusive repertoires and two have significantly overlapping repertoires. Mutational trajectories link one of those clusters to the clusters exclusive to BM and spleen, respectively. Thus, switched B cell memory is maintained in shared and exclusive compartments in a secondary lymphoid organ, i.e., the spleen, and in the BM, which harbors an exclusive population of quiescent, affinity-matured Bsm.

## Results

### Bsm are abundant in spleen and bone marrow.
Enumeration of CD19$^+$CD38$^+$CD138$^-$GL7$^-$ memory B cells expressing IgA, IgG1, or IgG2b, i.e., switched memory B cells, in spleen, lymph nodes, BM, Peyer's patches, and blood of individual mice, revealed that despite a large variability in total cell numbers, most Bsm were located in spleen, BM, and lymph nodes (Table 1, Supplementary Fig. 1a–d). In immunized C57BL/6 mice, kept under specific pathogen-free conditions, and in mice obtained from local pet shops, the spleen contained two to three times more Bsm than the BM. In these immunized C57BL/6 mice and pet shop mice, 18–41% of switched Bsm were located in the BM, 9–14% in peripheral lymph nodes and 32–60% in the spleen (Supplementary Fig. 1c, d). Remarkably, the spleens of feral mice (wild mice) were considerably smaller than those of C57BL/6 mice and pet shop mice (Supplementary Fig. 1e) as has been

previously reported for feral *M. musculus domesticus*[7]. These mice had about equal numbers of Bsm in BM and spleen. While these numbers reflect steady-state conditions in pet shop mice and feral mice, following s.c. injection of NP-KLH with LPS as an adjuvant, in C57BL/6 mice the vast majority of NP-specific Bsm were located in the BM more than 420 days after immunization (Fig. 1a), demonstrating that in this particular immune reaction, Bsm can be preferentially maintained in the BM.

### Bsm of the bone marrow are resting in G$_0$ of cell cycle.
Most Bsm of BM and spleen are resting in terms of proliferation, according to staining with Ki-67 (Fig. 1b). Ki-67 was expressed by no more than 9.3% (median: 8.0%) of Bsm in spleen, and 4% (median: 2.6%) of Bsm of BM, i.e., more than 90% of cells in the spleen and more than 95% in the BM were in the G$_0$ phase of cell cycle. Accordingly, Bsm of both spleen and BM were refractory to treatment with cyclophosphamide (Fig. 1c), a drug which eliminates proliferating cells. In contrast, the numbers of total CD19$^+$ B cells of the spleen and BM were significantly reduced after one week of cyclophospamide treatment.

### Bone marrow Bsm co-localize with VCAM-1$^+$ stromal cells.
In histological sections of the BM of Blimp1-green fluorescent protein (GFP) C57BL/6mice, we identified IgG2b$^+$ Bsm as IgG2b$^+$ IgD$^-$Ki-67$^-$ nucleated cells. IgG2b$^+$ plasma cells were excluded from analysis according to expression of GFP under control of the *Prdm1* (Blimp1) promoter (Fig. 1d, Supplementary Fig. 1f). IgG2b$^+$ Bsm were dispersed as single cells throughout the BM (Fig. 1d). In histological sections 75% of IgG2b$^+$ Bsm were observed in direct contact with cells expressing VCAM-1 and fibronectin (Fig. 1e, f), and a further 15–20% of Bsm within 10 μm vicinity of such stromal cells (Fig. 1f). 53% of the Bsm were directly contacting laminin-expressing stromal cells, and another 26% were in the 10 μm vicinity of such cells (Fig. 1f). Contact of IgG2b$^+$ Bsm to VCAM-1$^+$ stromal cells is deterministic, since it is significantly different from random association between the two cell types, as determined by simulation of random co-localization (Supplementary Fig. 1g)[8]. The co-localization of Bsm and stromal cells is in line with expression of VLA4 (CD49d/CD29), a receptor for fibronectin and VCAM-1, and VLA6 (CD49f/CD29), a receptor for laminin[9], by Bsm (Fig. 1g, CD19 staining and cell size shown in Supplementary Fig. 1h). About 10% of Bsm were in direct contact and 26% within 10 μm vicinity of cadherin 17 (Cdh17)-expressing stromal cells (Fig. 1f).

Taken together, Bsm are abundant in BM and spleen, where they rest in terms of proliferation. In the BM, Bsm are individually docked onto stromal cells.

### Bsm of bone marrow and spleen have distinct Ig repertoires.
Comparing the BCR repertoires of Bsm of spleen and BM of individual mice on the level of complementarity-determining region 3 (CDR3) of their immunoglobulin heavy chains, revealed only marginal overlap of CDR3 repertoires between Bsm expressing the same isotype residing in the spleen or BM of individual mice. This is shown in Fig. 2 and Supplementary Fig. 2 for IgG1/2$^+$ and IgA$^+$ Bsm of three individual C57BL/6J mice, which were immunized three times with NP-CGG. Biological and technical replicates served to determine how representative the samples were, and to control reproducibility (Supplementary Fig. 2a). Cosine similarity, a measure to determine the similarity of two groups irrespective of size, was significantly higher for biological replicates (0.65–0.97) than between samples from spleen and BM of each mouse (cosine similarity ~0.4) (Supplementary Fig. 2b). Overall, Bsm of BM and spleen show a similar clonal diversity and distribution of clonotype frequencies

**Table 1 Tissue distribution of isotype-switched memory B cells.**

| Hygiene status | Isotype | Spleen | Bone marrow | Peripheral LN | Mesenteric LN | Peyer's patches | Blood |
|---|---|---|---|---|---|---|---|
| C57BL/6 SPF | IgA | 1976 ± 455 | 35,634 ± 8754 | 4304 ± 1398 | 9090 ± 3290 | 691 ± 560 | 53,033 ± 11,414 |
| | IgG1 | 392 ± 166 | 24,390 ± 9098 | 9995 ± 2127 | 10,774 ± 3760 | 3737 ± 1363 | 63,317 ± 21,266 |
| | IgG2b | 2113 ± 580 | 106,280 ± 21,839 | 18,960 ± 9382 | 44,472 ± 9207 | 9287 ± 6027 | 283,616 ± 43,693 |
| pet shop | IgA | 2130 ± 590 | 77,630 ± 39,883 | 8907 ± 1996 | 26,300 ± 17,838 | 13,013 ± 6946 | 61,850 ± 18,233 |
| | IgG1 | 2394 ± 806 | 148,577 ± 81,700 | 36,323 ± 10,256 | 58,646 ± 17,547 | 17,609 ± 6470 | 222,179 ± 59,635 |
| | IgG2b | 9827 ± 2565 | 425,361 ± 223,940 | 147,720 ± 25,912 | 221,403 ± 42,907 | 136,631 ± 44,282 | 1,385,442 ± 390,533 |
| feral | IgA | 2169 ± 952 | 12,904 ± 3853 | 340 ± 202 | 3602 ± 733 | 3758 ± 1624 | 12,873 ± 5428 |
| | IgG1 | 2877 ± 713 | 38,719 ± 12,991 | 797 ± 348 | 53,537 ± 19,914 | 2305 ± 784 | 55,717 ± 19,544 |
| | IgG2b | 7373 ± 1882 | 73,974 ± 27,089 | 1321 ± 467 | 75,657 ± 28,597 | 4503 ± 1070 | 104,652 ± 37,075 |

Mean values (±SEM) of IgG1+, IgG2b+ and IgA+ CD19+CD138−CD38+CD11c−GL7−IgM−IgD− small lymphocytes; total cell numbers as calculated per organ. N = 42 (from eight experiments with five different immunizations protocols performed in C57BL/6 mice aged 4–20 months and held under SPF conditions), 13 (feral) and 13 (pet shop).

(Supplementary Fig. 2c, d), with the samples representing more than 97% of the predicted entire repertoire each (Supplementary Fig. 3 and Supplementary Table 1). To discriminate between stochastic and directed repertoire overlap, we simulated random overlap between two biological samples by randomly reshuffling the sequences observed, resulting in significantly ($P < 0.001$) higher overlaps (Fig. 2b, d) than those observed (Fig. 2a, c). Remarkably, several of the most frequently expressed Ig heavy chain V-genes were found predominantly either in the spleen or in the BM, e.g., IGHV1-5 of mice 1 and 3 (Fig. 2e). The segregation of repertoires of Bsm of spleen and BM strongly argues for a tissue-specific compartmentalization of Bsm.

**Transcriptional heterogeneity of Bsm in bone marrow and spleen.** In order to resolve the heterogeneity of Bsm within as well as between spleen and BM, we analyzed single-cell transcriptomes of Bsm isolated from both organs of two individual mice. Bsm from C57BL/6J mice immunized three times with NP-CGG/incomplete Freund's adjuvant (IFA) i.p. 60 days prior to analysis were enriched by MACS with CD19 microbeads (Miltenyi Biotech) and isolated by FACS as IgG1+IgG2b+CD19+CD38+GL7−CD138−IgM−IgD− cells. In total, 6047 Bsm from the spleen and 4164 from the BM were subsequently sequenced using 10× genomics-based droplet sequencing. An average of 42,407 reads per cell defined a median number of 1629 transcribed genes per cell, corresponding to a sequencing saturation of approximately 70%. For 4754 Bsm from spleen and 2947 from BM, we also determined the sequences of their antibody heavy and light chains. In line with the isolation protocol and degree of transcriptome saturation, most cells expressed the B cell marker genes *Cd19* (86% of all cells), *Cd38* (36% of all cells), *Pax5* (38% of all cells), and *Ptprc* (CD45, 80% of all cells) (Supplementary Fig. 4a). Transcription of *Sdc1* (CD138, <0.1%, a surface protein expressed on plasma blasts and plasma cells) and *Bcl6* (2.5%, a transcription factor characteristic for germinal center B cells) was not detectable overall (Supplementary Fig. 4a). According to their transcriptomes, Bsm robustly clustered into six different populations (Fig. 3a). Dimensionality reduction by Uniform Manifold Approximation and Projection (UMAP)[10] was similar with regard to cluster separation compared to *t*-distributed Stochastic Neighbor Embedding (tSNE) plots (Supplementary Fig. 5). Bsm of clusters I, II, III, and VI were present in both the spleen and BM of the two mice at varying frequencies. Bsm of cluster IV were almost exclusively located in the spleen, while Bsm of cluster V were virtually exclusively located in the BM (Fig. 3b). While clusters III (91% IgG1, 9% IgG2b/c) and V (94% IgG1, 6% IgG2b/c) were predominantly cells of IgG1 isotype, cluster VI was highly enriched for cells of IgG2b/2c isotype (4% IgG1, 96% IgG2b/c).

Cluster I (84% IgG1, 16% IgG2b/c), II (36% IgG1, 64% IgG2b/c), and IV (71% IgG1, 29% IgG2b/c) consisted of cells of mixed isotypes (Fig. 3c).

Signature genes characteristic for the different clusters are displayed in Fig. 3d. These 45 genes represent the top predictive genes according to sensitivity and specificity (receiver-operator characteristics, ROC), with an area under curve (AUC) of >0.7. Bsm of Cluster I were characterized by low levels of *Cr2* (CD21) (14% of cells in cluster 1) and intermediate levels of *Fcer2a* (CD23) (52% of cluster I cells) (Fig. 3d, e). 95% expressed the transcription factor *Krueppel-like factor 2* (*Klf2*), 93% the cytoskeletal protein *Vimentin* (*Vim*)[11] and 62% the golgi/endosomal-associated gene *Prostate androgen-regulated mucin-like protein 1* (*Parm1*)[12]. In summary, Bsm of cluster I resemble transitional memory B cells[13,14]. 63% of Bsm of cluster II expressed high level of *Fcer2a* and intermediate levels of *Cr2* (24% of cluster II Bsm). Cells of cluster II also expressed the complement decay-accelerating factor *Cd55*[15] in 75% of the cases and 77% expressed the transcription factor *Foxp1* (*Forkhead Box P1*)[16]. This gene expression pattern suggests that Bsm of cluster II are follicular memory B cells[13]. Bsm of cluster III resemble age-associated memory B cells (ABC), in that they expressed low levels of *Cr2* (3%) and *Fcer2a* (15%). We detected high levels of the integrin gene *Itga4* (CD49d) in 91%, *Itgam* (CD11b) in 4% (Supplementary Fig. 4b) and the chemokine receptor gene *CXCR3* in 47% of cluster III Bsm[17]. In addition, 62% of cluster III Bsm expressed the gene encoding the calcium binding protein *S100a6*[18]. However, they lacked the expression of *Itgax*, the gene encoding for CD11c (Supplementary Fig. 4b), which has been associated with ABC (reviewed by Rubtsova et al.[17]). Bsm of cluster IV, residing exclusively in the spleen, resemble marginal zone B cells in that 53% of them expressed high levels of *Cr2* and all of them the gene *Cd52*, and 59% expressed intermediate levels of *Fcer2a*[13,19]. Cluster V, found exclusively in the BM, is comprised of Bsm expressing low levels of *Cr2*, and *Fcer2a* in 6 and 24% of Cluster V Bsm, respectively. In contrast, they expressed relatively high levels of various genes encoding for ribosomal proteins, such as *Rps26-ps1* (76% of cluster V Bsm). 10% of cluster V Bsm also expressed the gene *Cxcr4*, encoding a chemokine receptor associated with BM homing[20]. Bsm of cluster VI resemble B1 cells, in that they are characterized by expression of the scavenger receptor *Cd5*[21] in 8% (Supplementary Fig. 4b), the Protein tyrosine phosphatase 22 gene *Ptpn22* associated with hypo-responsiveness of B cells in the context of autoimmunity[22] in 56% and the gene encoding the tetraspanin *Cd9* expressed on regulatory B cells[23] in 49% of the cells.

The differential expression of *Cr2* (CD21/35) and *Fcer2a* (CD23) by Bsm subpopulations was also evident on the protein level

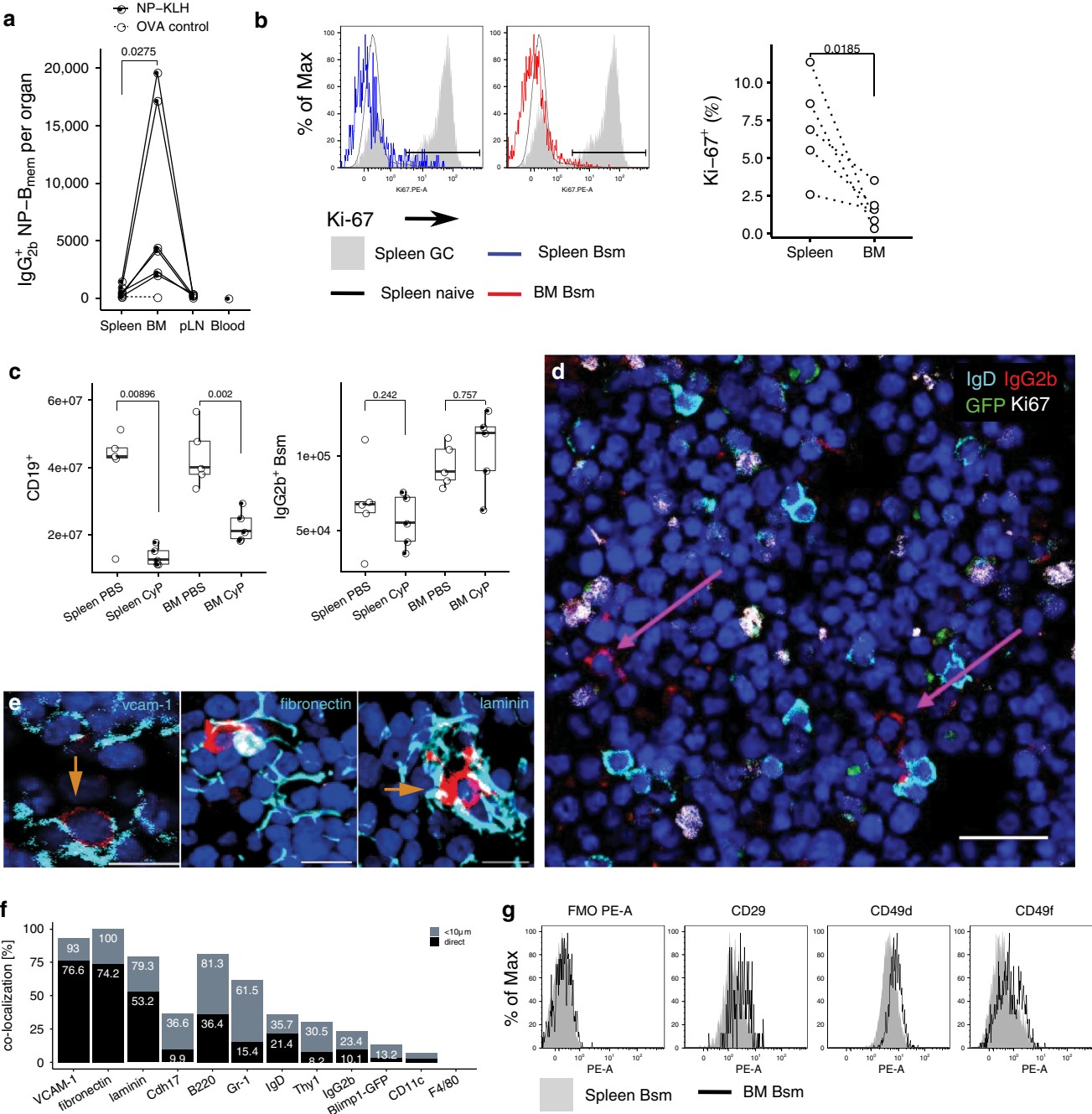

(Fig. 4). By flow cytometry, and using only CD21/35 and CD23 on gated Bsm, we were able to identify populations of CD21/35$^{intermediate}$/CD23$^{intermediate}$ (cluster I), CD21/35$^{intermediate}$ CD23$^{high}$ (cluster II), and CD21/35$^{high}$CD23$^{low}$ (cluster IV) cells in both the spleen and BM (Fig. 4a, b). Cells of clusters III, V, and VI were among the CD21/35$^{low}$CD23$^{low}$ population. Within this population, cells from cluster VI could be discriminated from clusters III and V by their expression of CD5 (CD21/35$^{low}$CD23$^{low}$/ CD5$^+$) (Fig. 4a–c). Although cells from both clusters III and V expressed *Itga4* (CD49d), notably CD49d expression was higher in the BM than the spleen (Fig. 1g), the presence of a population expressing higher levels of CD49d within CD21/35$^{low}$CD23$^{low}$/ CD5$^-$ cells points to an enrichment of cluster III in this population (Fig. 4a–c). The accumulation of CXCR3 and CD11b expressing Bsm within the CD21/35$^{low}$CD23$^{low}$/CD5$^-$/CD49d$^{high}$ cells further

supports that clusters III and V can be separated by CD49d expression (Fig. 4a–c).

In addition to the specific gene expression signatures (Fig. 3), Bsm of the clusters differed in the expression of gene sets indicating physiological activitiy. Single cell gene set enrichment analysis of expression of the Reactome[24] and Gene Ontology (GO)[25–27] gene sets revealed that Bsm of both organs express gene sets associated with oxidative phosphorylation (Reactome: TCA/electron transport) and integrin signaling (Reactome: integrin cell surface interactions) (Supplementary Fig. 4c), but not gene sets associated with glycolysis or RNA polymerase II-mediated transcription. Confirming their quiescence, Bsm of cluster V, located exclusively in the BM, did not express gene sets associated with BCR signaling, DNA synthesis, or regulation of cell cycle progression. Conversely, many Bsm of cluster IV,

**Fig. 1 The bone marrow contains a major population of isotype-switched non-proliferating memory B cells. a** Quantification of NP-specific IgG2a/b$^+$ spleen, peripheral lymph nodes, blood and bone marrow (BM) memory B cells. Female C57BL/6 mice immunized with NP-KLH/LPS SC. Numbers of NP-binding IgG2b$^+$ cells in Spleen, BM, blood, and peripheral lymph nodes (pLN) determined by flow-cytometry on d421 or d426 post immunization; pooled from two independent experiments. OVA ctrl: staining controls from mice immunized with the irrelevant antigen ovalbumin (OVA). Gated for IgG2b$^+$CD19$^+$ CD38$^+$CD138$^-$GL7$^-$CD11c$^-$IgM$^-$IgD$^-$PI$^-$ small lymphocytes (cf. Supplementary Fig. 5). Lines connect samples from one individual, paired one-sided $t$-test for spleen and BM samples. **b** Flow-cytometric quantification of Ki-67 expression in IgG2b$^+$ Bsm (IgG2b$^+$CD19$^+$CD38$^+$CD138$^-$GL7$^-$CD11c$^-$IgM$^-$IgD$^-$ PI$^-$ small lymphocytes) splenic naïve (IgM$^+$IgD$^+$IgG2b$^-$CD19$^+$CD38$^+$CD138$^-$GL7$^-$CD11c$^-$ PI$^-$ small lymphocytes) and germinal center (GC) (CD19$^+$ CD38$^{lo}$GL7$^+$CD11c$^-$PI$^-$ lymphocytes) B cells. Frequencies of Ki-67$^+$ cells within the subset, data in right graph from two independent experiments using pooled cells from 4 to 20 C57BL/6 mice, paired one-sided $t$-test. **c** Flow-cytometric quantification of CD19$^+$ B cells and IgG2b$^+$ memory B cells in female C57BL/6J mice treated with Cyclophosphamide (CyP) or untreated controls (PBS) after immunization with 3× NP-CGG/IFA. Analysis performed after 7 days of CyP. IgG2b$^+$ B cells quantified as IgG2b$^+$CD19$^+$CD38$^+$CD138$^-$GL7$^-$CD11c$^-$IgM$^-$IgD$^-$PI$^-$ small lymphocytes, CD19$^+$ B cells as CD19$^+$CD138$^-$ PI$^-$ lymphocytes (Welch's test, one-sided). Representative data for one of two independent experiments ($n = 5$ per group). Boxplot indicates median, first and third quartiles, whiskers: 1.5 IQR. **d** IgG2b$^+$ B memory cells (Ki-67$^-$ IgD$^-$ Blimp1$^-$GFP$^-$) are dispersed as single cells throughout the BM. Arrows indicate IgG2b$^+$DAPI$^+$ cells. Scale bar: 20 μm. Micrograph representative of five slides from two female C57BL/6 mice. **e** Co-localization of IgG2b$^+$ GFP$^-$IgD$^-$IgG2b$^+$ cells (arrows) with mesenchymal stromal cells. Arrows indicate IgG2b$^+$DAPI$^+$ cells. Representative micrograph. Scale bars: 10 μm. **f** Co-localization of IgG2b$^+$ cells to mesenchymal stromal cells. Graph shows frequency of IgG2b$^+$ cells in direct contact (black) or within 10 μm (gray) of a cell stained for the respective molecule. **g** Flow cytometric quantification of surface expression of the VLA-4 and VLA-6 components CD29, CD49d, CD49f in spleen and BM IgG2b$^+$ Bsm. Gated for IgG2b$^+$CD19$^+$CD38$^+$CD138$^-$GL7$^-$CD11c$^-$IgM$^-$IgD$^-$PI$^-$ small lymphocytes, histogram plots representative of three biological replicates from C57BL/6 females. Source data for Fig. 1a–c, f are provided as a Source Data file.

located only in the spleen, expressed these gene sets. In addition, these Bsm exclusively expressed a gene set indicating cognate activation. Bsm of cluster III were exclusively enriched for a gene set associated with lymphocyte migration (GO term lymphocyte migration).

**BCR repertoires and trajectories of Bsm clusters.** Comparison of the BCR repertoires (paired Ig heavy and light chain) of the Bsm clusters I, II, and IV showed exclusive repertoires, as determined by simulation of random distribution of BCR sequences to the different clusters (Fig. 5a, Supplementary Table 1), confirming the results obtained for other mice and the bulk analysis of their Bsm of spleen and BM (Fig. 2 and Supplementary Fig. 2). A significant overlap of repertoires was found between Bsm of cluster VI from the spleen and BM. The same was true for Bsm of clusters III and V, which showed a significant overlap of their BCR repertoires between clusters as well as between those of spleen and BM. Together these results imply that those Bsm clusters were interconnected in their generation and/or maintenance.

Bsm with a high-degree of somatic hyper-mutation (>1% mutation in framework (FR)1-3) were more frequent in clusters I, III, IV, and V (Fig. 5b) than in clusters: II and VI. Interestingly, the genes encoding for CD80 (*Cd80*) and CD273 (*Pdcd1lg2*), which have been described to correlate with an increase in somatic hypermutations[28] were significantly enriched in those clusters (Chi-squared with Yates' correction comparing distribution of either CD80 or CD273 expressing cells in clusters I, III, IV, and V as well as in cluster II and VI, significance level of $P < 0.01$). In contrast, the number of *Nt5e* (CD73) expressing cells was reduced in clusters I, III, IV, and V, as compared to random distribution of transcription (Fig. 5b). Trajectories based on somatic hyper-mutation of BCR heavy and light chain identified cluster III from which clones with higher mutation counts would have originated. A total of 187 significant clones (significantly over-represented compared to randomization) were observed in other clusters originating from either spleen or BM cluster III. The frequencies of transitions, i.e., presence in more than one cluster, are indicated in Fig. 5c, taking into account the population size of each cluster in each organ. A trajectory was considered significant if a $P$-value < 0.01 was observed (indicated in red, Fig. 5a, c), compared to random distribution of mutated BCR clones into different clusters. Figure 5d indicates the direction of mutational trajectories, and the number of clones

involved. Of the 187 clones of cluster III, 55 clones were represented with additional mutations in the very same cluster of the other organ. Forty-five of the cluster III clones from spleen or BM were represented with additional mutations in cluster V of the BM, 23 clones were among the rare cells of cluster V of the spleen. Forty clones of either spleen or BM cluster III were also identified in cluster I of the BM, with additional mutations. There was a significant exchange between splenic and BM cluster V (14 clones) as well as a minimal but significant exchange between the splenic and BM cluster VI with either splenic or BM cluster II (8 clones). Additionally, we were able to detect a significant exchange between BM cluster III and the spleen-exclusive cluster IV (24 clones), as well as between splenic and BM cluster III and splenic cluster V (23 clones).

In summary, mutational trajectories link cluster II to cluster VI and cluster III to cluster V, and to lesser degree to cluster I. Together these data suggest that Bsm of cluster III of spleen and BM represent one interconnected compartment.

## Discussion

In secondary immune responses, memory B lymphocytes provide rapid immune protection, with antibodies of enhanced affinity and adapted function, which is reflected by their switched isotype. While IgM memory B cells have been shown to contribute essentially to germinal center reactions in secondary immune responses, Bsm have been considered to be primarily direct precursors of plasma cells[29–33]. The generation of memory B lymphocytes and their role in secondary immune responses has been subject of intense research, but little is known about their maintenance over time in the apparent absence of antigen. For memory T lymphocytes, recent evidence has unveiled a highly structured organization of their maintenance, describing populations of circulating and tissue-resident memory T cells (Trm)[3]. Trm have been described for nearly all organs investigated, including the BM[8,34,35]. Tissue-residency has been demonstrated by parabiosis, solid organ transplantation, label perfusion[36] but above all, by exclusive maintenance of antigen-receptor repertoires in defined tissues[34,37]. While exclusive repertoires define resident lymphocyte populations, besides recirculation, different resident populations could also share repertoires, if they were linked in generation.

Memory B lymphocytes expressing antibodies of switched isotype can be detected in many organs, by and large with a similar phenotype and with tetanus-specific Bsm in all organs,

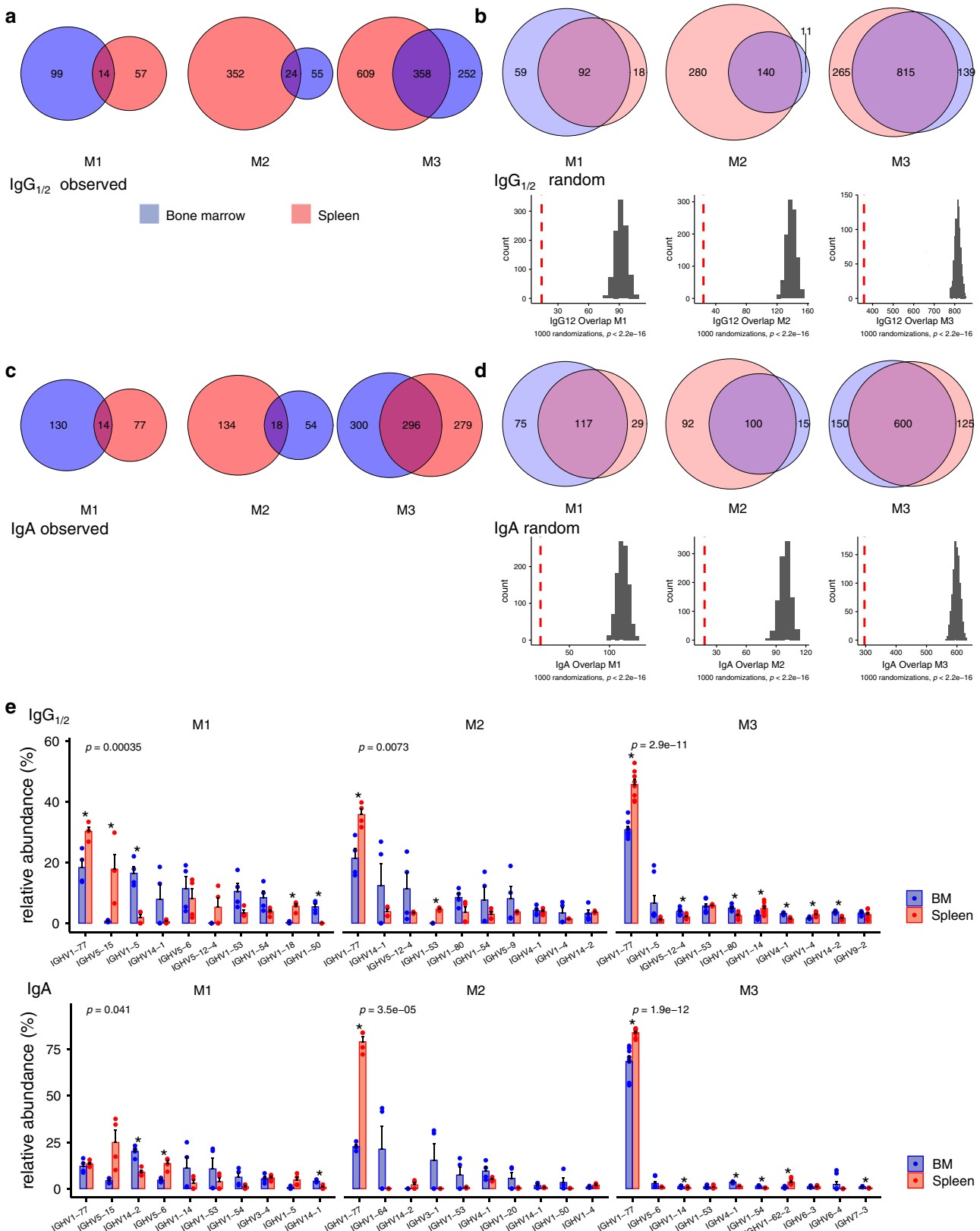

arguing that Bsm form one population of interconnected eventually circulating memory cells[4]. In contrast, Mamani-Matsuda and colleagues had shown that *Vaccinia*-specific Bsm are mainly located in the human spleen and not in the BM, as far as analyzed[6]. Recently Allie and colleagues have demonstrated the existence of resident Bsm in murine lungs[38]. Here

we show that BM hosts a prominent population of memory B lymphocytes, in particular in feral mice. This is in line with earlier evidence, that in humans, IgM memory B cells can migrate to the BM[39]. Comparison of Bsm in the spleen and BM of individual mice, on the level of single cell transcriptomes and antibody repertoires, demonstrates that Bsm of both

**Fig. 2 Spleen and bone marrow isotype-switched memory B cells are distinct in Ig heavy chain repertoire. a, c** Observed overlap between the IgG1/2[+] or IgA[+] heavy chain CDR3 repertoire of switched memory B cells from spleen and bone marrow (BM) or **b**, **d** random distribution (upper panels). Venn diagrams represent clonotype presence in a given sample: numbers indicate clonotypes present in one organ exclusively or in both (overlap). Random distributions (**b**, **d**) show median values for 1000 random distributions drawn while retaining the number of clones initially observed in the sample. Histograms (**b**, **d**, lower panels) represent number of clonotypes in spleen and BM in 1000 randomized distributions of observed clonotypes to spleen and BM, dashed red line indicates experimentally observed number of clonotypes present in both spleen and BM. P value of one-sided t-test for difference of randomized overlap against observed. **e** VH gene recruitment to spleen and BM IgG1/2[+] (upper panel) and IgA[+] (lower panel) memory B cells, represented as frequency of a particular VH gene among total CDR3s per organ. Bars show relative abundance of the ten most abundant VH genes, error bars indicate SEM. Significance of difference in VH gene distribution to Spleen and BM assessed by MANOVA, P values corrected for multiple testing (Benjamini-Hochberg), * indicates significant difference in means for a particular VH gene (Welch's test, two-sided). M1–M3: replicate samples of three female C57BL/6 mice immunized 3× NP-CGG/IFA. Only clones consistently found in technical replicates were considered. Source data for Fig. 2a–e are provided as a Source Data file.

organs comprise different compartments, and defines their heterogeneity.

By comparing the antibody repertoires of Bsm of spleen and BM of individual mice, we show that both organs host prominent populations of exclusive, i.e., resident and tissue-specific, as well as smaller populations of connected Bsm, with overlapping repertoires. Transcriptomes of individual Bsm from murine spleen and BM define six clusters, four of which are present in both, spleen and BM. One is only found in spleen and another one only in the BM.

Cluster I, a major population of Bsm in both, spleen and BM, consists of Bsm expressing low levels of CD21 and intermediate levels of CD23. This combination is characteristic for transitional, immature B cells (T1/2), which are believed to represent a developmental stage of negative selection against autoreactivity in BM (T1) and spleen (T2)[13]. Classical transitional B lymphocytes express IgM and IgD, and so far[40], isotype-switched transitional B cells have not been described. However, the antigen-receptor repertoires of Bsm of cluster I of spleen and Bm differ, as well as being distinct from those of all other Bsm, indicating that Bsm of cluster I are resident cells, selected in exclusive immune reactions. It should be noted that those repertoires show a high degree of somatic hyper-mutation, indicating an extensive selection process.

Bsm of cluster II express intermediate levels of CD21 and high levels of CD23, a pattern reminiscent of follicular B cells, generated in germinal center reactions[13]. Their antigen receptor repertoires are different between spleen and BM, and also different from all other Bsm. Again indicating that Bsm cluster II in BM and spleen are resident cells, generated in different immune reactions and most likely in germinal centers[41]. While their antibodies show somatic hyper-mutation, its extent is lower than that of most other Bsm, except those of cluster VI.

Cluster III consists of Bsm expressing low levels of CD21 and CD23, and accumulated cells expressing CD11b and CXCR3, all markers of "aged" memory B cells[17]. T-bet, another marker gene of IgG Bsm[42–45], in particular of "aged" memory B cells, was not quantitatively detectable in the present analysis either due to the sequencing depth or low expression of this transcription factor per cell. "Aged" Bsm have been described to be generated in germinal centers by combined activation of B cells through TLR and antigen receptors[42–44].

Bsm of cluster III express highly mutated antibodies, and their repertoires overlap significantly between spleen and BM. Moreover, their repertoires also overlap with Bsm of cluster V in BM, and the few cells of cluster V in the spleen. Interestingly, mutational trajectories indicate that Bsm of cluster III are the precursors of those of cluster V. Mutational trajectories also link Bsm of cluster III of spleen and BM, in both directions, indicating that those clusters either would have been connected during establishment, and/or are still connected during maintenance of memory, i.e., represent circulating Bsm. In line with this, Bsm of cluster III are exclusively expressing a gene set associated with lymphocyte migration.

Bsm of cluster IV are found almost exclusively in the spleen. They express high levels of CD21 and intermediate levels of CD23. This, together with their transcriptional signature classifies them as marginal zone-like memory B cells[13,19]. Bsm of marginal zones have been described before[46]. Here, we show that their antigen receptors are highly mutated and that their repertoire is different from the repertoire of Bsm of all other clusters, indicating that they are a resident population, generated in exclusive immune reactions.

An exclusive BM population of Bsm is seen in cluster V. Bsm of cluster V express low levels of CD21 and CD23. They are quiescent, as reflected by their lack of expression of gene sets associated with BCR receptor signaling, DNA synthesis or regulation of cell cycle progression. In relation to other genes, they express high levels of housekeeping genes encoding ribosomal proteins[47]. Their antigen receptor repertoire is different from that of Bsm of clusters I, II, IV, and VI, but shows considerable overlap with Bsm of cluster III (see above), which according to mutational trajectories, qualify as their precursor cells. Interestingly, different from Bsm of cluster III, Bsm of cluster V, however, do not express genes associated with migration.

Finally, Bsm of cluster VI, present in both spleen and BM, are exclusively IgG2 expressing cells. They express CD5 and CD9, markers of cells of the B1 lineage[21]. Repertoires of Bsm from spleen and BM overlap significantly, showing that these populations have been connected in development. It should be noted, however, that Bsm of cluster VI do not express genes associated with lymphocyte migration in the maintenance phase of memory.

In summary, the clustering of Bsm according to their transcriptomes reveals an unforeseen heterogeneity, with populations so far not described (clusters I, V), or poorly characterized (cluster III, IV). The vast majority of these cells, those of clusters I, II, IV, V, and VI are apparently permanent residents of their tissue, since they have exclusive antigen receptor repertoires and/or exclusive location, and they do not express genes associated with lymphocyte migration. 10–20% of Bsm of spleen and BM, those of cluster III, resemble "aged" memory B cells and qualify as circulating memory B cells, with largely overlapping repertoires of spleen and BM, and expression of genes associated with migration. Thus, as described for memory T lymphocytes[3], a significant proportion of Bsm of BM are resident. Whether they convey preferentially long-term memory to systemic challenges, like their T cell counterparts[37], remains to be shown.

Moreover, Bsm are apparently maintained in the BM like memory T and memory plasma cells[3], where they individually dock onto mesenchymal stromal cells, expressing VCAM1, which provide a niche for their maintenance. We did not observe a preferential colocalization to Cadherin-17 expressing stromal cells

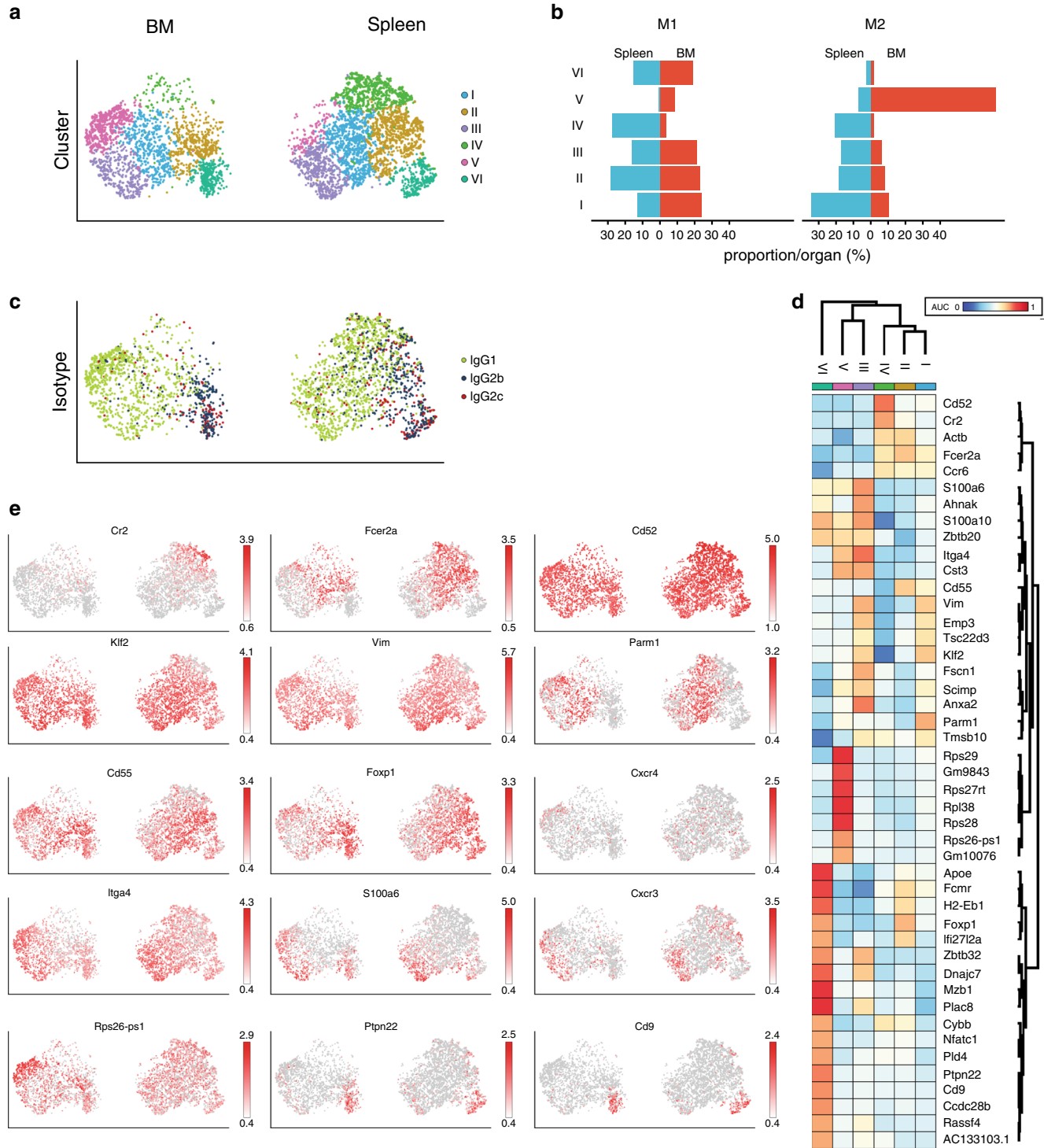

**Fig. 3 Heterogeneity of switched memory B cells is differentially represented in spleen and bone marrow.** Cells for single cell sequencing were FACSorted as IgG-expressing CD19+CD38+CD138-GL7- small lymphocytes. **a** Six transcriptionally defined clusters were identified by shared nearest neighbor (SNN) modularity optimization based clustering algorithm mapped to tSNE representation of spleen and bone marrow (BM) cells. tSNE coordinates and clustering was computed for 4754 from spleen and 2947 from BM cells, presentation is separated by organ. **b** Percentage of cells per cluster in each organ by mouse. **c** Distribution of IgG subclass mapped on tSNE. **d** Signature genes for each cluster, area under curve (AUC) of receiver-operator characteristics (ROC) of >0.7. **e** Distribution of transcription levels for representative genes mapped on tSNE. Cells for single cell sequencing were FACSorted as IgG-expressing CD19+CD38+CD138−GL7− small lymphocytes from female C57BL/6 mice. Source data are provided as a Source Data file.

of the BM, a colocalization that has been reported relevant for Bsm of the spleen[48]. Instead, we observed a co-localization of Bsm of the BM to laminin-expressing stromal cells. This is in striking homology to IgG-secreting memory plasma cells, which

in the BM, but not in the spleen, require laminin-beta1 for their maintenance[49].

Resident Bsm of secondary lymphoid organs, e.g., the spleen, are able to participate in secondary immune reactions, inside or

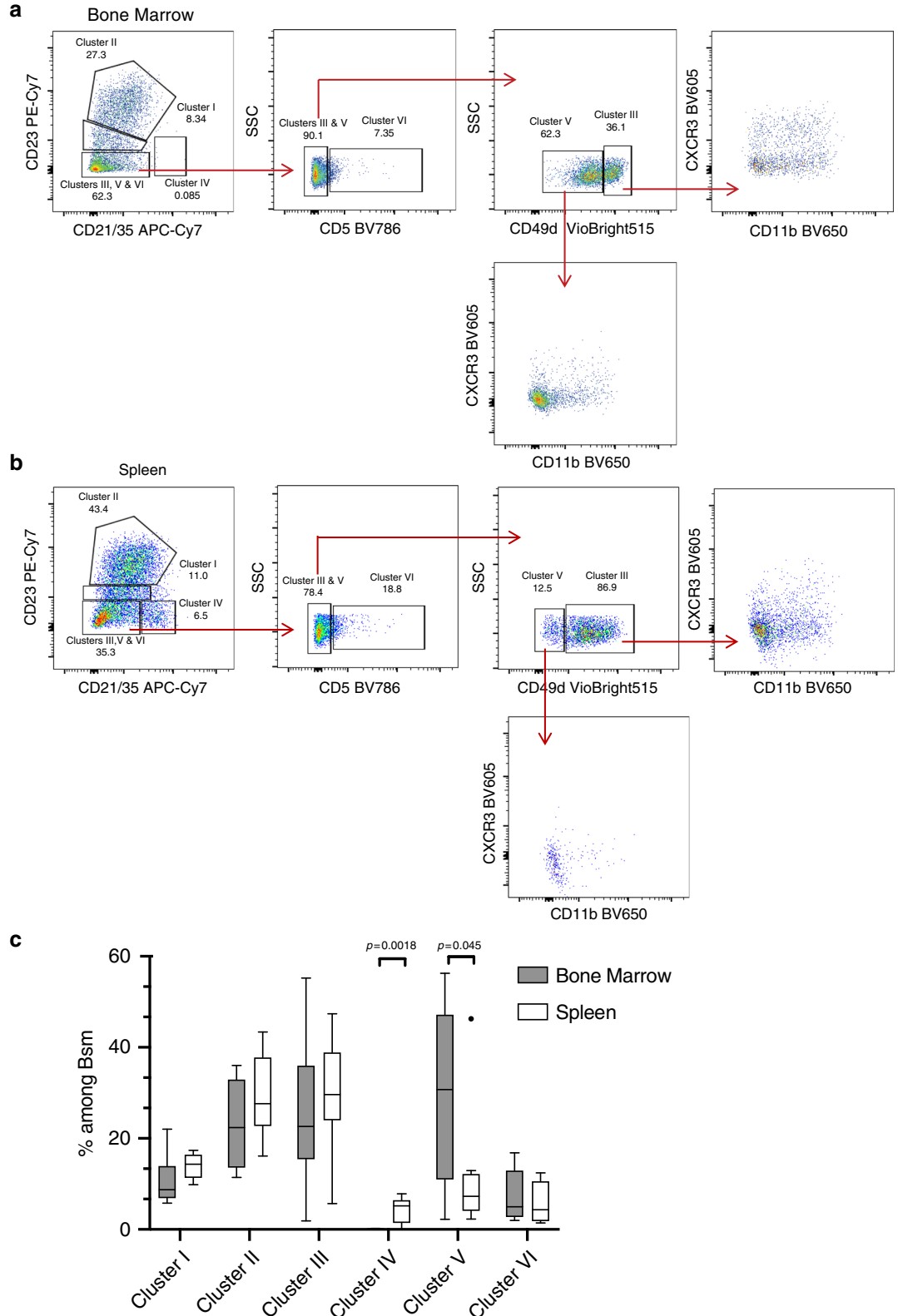

outside[50,51] of germinal centers. However, the raison d'être of resident Bsm of the BM is less clear at first glance. For resident memory T cells of the BM, we have demonstrated their effective reactivation in the BM in immune clusters[52] consisting of reactivated memory T cells and antigen-presenting cells, including B cells[53]. Whether or not Bsm of the BM are reactivated in a similar way, remains to be shown. Assuming a similar reactivation, this

would be an efficient and fast way to secure rapid generation of new (memory) plasma cells and secreted antibodies against blood-borne antigens in situ, to provide enhanced local[38] and systemic protection. In line with this, cells of clusters I, III, IV, and V express the highest rates of somatic hypermutation, and they express CD80 and PD-L2 which have been associated with high somatic hypermutation and capacity for direct

**Fig. 4 Surface staining of CD21/35, CD23, CD5, and CD49d resolves six distinct populations of switched memory B cells in spleen and bone marrow.**
**a, b** Flow-cytometric identification of distinct populations among IgG⁺ memory B cells gated as IgG1⁺/IgG2b⁺ CD19⁺IgM⁻IgD⁻CD138⁻GL7⁻CD93⁻ live small lymphocytes (cf. Supplementary Fig. 5) in bone marrow (BM) (**a**) and spleen (**b**) of female C57BL/6 mice. CD21/35 and CD23 resolve three subsets resembling clusters I, II, and IV as identified by transcriptional profiles. Expression of CD5 separates the CD21/35⁻CD23⁻ population to a subset resembling cluster VI while CD5⁻ cells are divided by high and intermediate staining of CD49d into two subsets, which differ in expression of CXCR3 and CD11b and resemble clusters III and V. Dotplots represent one of eight mice immunized 3× with NP-CGG/IFA. **c** Boxplots represent the frequency of subsets by cytometry according to expression of CD21/35, CD23, CD5, and CD49d among IgG⁺ Bsm. A non-parametric ANOVA (Friedman test) was performed (Two-stage linear step-up procedure of Benjamini, Krieger, and Ykutieli) $P < 0.0001$, followed by a one-sided paired $t$-test between spleen and BM cluster IV and V, respectively, $n = 8$. Boxplot indicates median, first and third quartiles, whiskers: 1.5 IQR. Source data for Fig. 4c is provided as a Source Data file.

differentiation into plasmablasts[29]. The contribution of BM-resident Bsm to secondary germinal center reactions in secondary lymphoid organs remains to be demonstrated, and it would require their mobilization and emigration from the BM. In steady state, only Bsm of cluster III have the transcriptional potential to do so.

## Methods

**Mice.** C57BL/6J/N mice were purchased from Charles River (Sulzfeld, Germany) or Janvier Labs (Le Genest-Saint-Isle, France). Mice expressing GFP under the control of the *Prdm1* promoter (Blimp1-GFP)[54] were bred at the DRFZ animal facility. C57BL/6 and Blimp1-GFP mice were housed under specific pathogen-free conditions. Experimental treatment and control groups were co-housed. Pet mice were obtained as adult animals at pet shops in Berlin. Feral mice were caught as free-living animals at non-residential farm buildings in Altlandsberg, Brandenburg, Germany. Live traps were set up and controlled by a trained veterinarian. Mice were anesthetized with isofluran prior to sacrifice. All animal experiments were performed according to institutional guidelines and licensed under German animal protection regulations by the Landesamt für Gesundheit und Soziales Berlin and Landesamt für Arbeitsschutz, Verbraucherschutz und Gesundheit Brandenburg.

**Immunizations and cyclophosphamide administration.** Mice aged 8–12 weeks were immunized s.c. with 100 µg NP-KLH 10 µg LPS (E. coli, InvivoGen). For boost immunizations 10 µg NP-KLH without adjuvant was used. Alternatively, mice aged 8–12 weeks were immunized three times with 100 µg NP-CGG in IFA i.p. at 21 days interval. For further quantification of Bsm in C57BL/6 laboratory animals, in two experiments mice were infected with $2 × 10^5$ plaque-forming units Armstrong strain of lymphocytic choriomeningitis virus i.p.[35] or, alternatively, infected intravenously with $10^6$ colony-forming units of attenuated *Salmonella enterica* serovar typhimurium strain SL7207[49].

Three times NP-CGG/ IFA-immunized (at 21 days interval) mice were i.v. administered for one week with either Cyclophoshamide (CyP, 50 mg/kg) or PBS starting 30 days after the last immunization. Analysis was performed on day 3 after the last CyP injection.

**Cytometry and cell sorting.** In order to stain lymphocytes for multicolor flow cytometry, cells were resuspended in PBS with 0.5% BSA and 5 µg per ml Fcγ receptor IIB-blocking antibody (DRFZ, clone 2.4G2) or 10 µl of FcR Blocking Reagent (Miltenyi) per 10⁷ cells at up to $5 × 10^8$ cells per ml. Cell separation by FACS was done at BD FACSAria II. For cytometric analyses Canto II, Fortessa, Symphony (BD) or MACSQuant (Miltenyi) machines were used. Data were analyzed by FlowJo (version 9, TreeStar).

Before staining, B cells were enriched by magnetic separation using anti-mouse CD19 microbeads (Miltenyi). Antibodies against following murine antigens were used: CD5 (53-7.3, BV786, BD OptiBuild Cat No 740842, 1:200), CD11b (M1/70, BV650, BioLegend Cat No 101239, 1:25), CD19 (1D3, PacB DRFZ, and 1D3, BUV395, BD Horizon Cat No 563557, 1:400), CD21/35 (7G6, BUV737, BioLegend, Cat No 62810, 1:400, and 7E9, APC-Cy7, BioLegend Cat No 123418, 1:400), CD23 (B3B4, PE-Cy7, BioLegend, Cat No 101614, 1:400), CD29 (HMß1-1, PE, Miltenyi Cat No 130-102-602, 1:25), CD38 (90, PE-cy7, BioLegend Cat No 102718, 1:400), CD49d (R1-2, PE, BioLegend Cat No 103608, 1:200 and REA807, VioBright515, Miltenyi Cat No 130-111-829, 1:200), CD49f (GoH3, PE, eBioscience Cat No 12-0495-82, 1:100), CD93 (REA298, PE, Miltenyi Cat No 130-104-157, 1:50, CD138 (281-2, PE, BD Biosciences Cat No 553714, 1:800), CXCR3 (CXCR3-173, BV605, BioLegend Cat No 126523, 1:23), GL7 (GL7, PE, DRFZ), IgA (C10-3, FITC, BD Biosciences Cat No 559354, 1:100), IgD (11.26c, PE, AF430, DRFZ and REA772, PE, Miltenyi Cat No 130-111-310, 1:100), IgG₁ (A85-1, FITC, BD Biosciences Cat No 553443, 1:1000, and X-56, VioBlue, Miltenyi Cat No 130-099-089, 1:50), IgG₂ₐ/ᵦ (R2-40, FITC 1:100, PE, 1:1000, BD Biosciences Cat No 553399 and X-57, VioBlue, Miltenyi Cat No 130-099-083, 1:50), IgG₂ᵦ (A95-1, FITC, BD Biosciences Cat No 553988, 1:800 and MRG2b-1, FITC, BioLegend Cat No 406706, 1:1000), IgM (M41, AF405, DRFZ and REA979, PE, Miltenyi Cat No 130-116-209, 1:100) and Ki-67 (B56, PE, BD Biosciences Cat No 556027, 1:400). Dead cells were excluded using PI staining or the Zombie Aqua Fixable Viability Kit (BioLegend). Flow cytometric

measurements and cell sorting were done according to standards defined in the guidelines to flow cytometry and cell sorting in immunological studies[55].

**Single cell RNA and single cell BCR repertoire sequencings.** Single-cell suspensions from BM of 3× NP-CGG immunized female C57BL/6 mice were prepared and CD19⁺ cells were enriched by magnetic cell sorting using anti-CD19 microbeads (Miltenyi Biotech). Ex vivo IgG1⁺/IgG2b⁺CD19⁺CD38⁺GL7⁻CD138⁻IgM⁻IgD⁻ memory B cells were isolated by FACS (Influx cell sorter (BD Bioscience)) and applied to the 10× Genomics platform using the Single Cell 5′ Library & Gel Bead Kit (10× Genomics) following the manufacturer's instructions. The amplified cDNA was used for simultaneous 5′ gene expression (GEX) and murine BCR library preparation. BCR transcripts were amplified by Chromium Single Cell V(D)J Enrichment Kit for murine B cells (10× Genomics). Upon adapter ligation and index PCR, the quality of the obtained cDNA library was assessed by Qubit quantification, Bioanalyzer fragment analysis (HS DNA Kit, Agilent) and KAPA library quantification qPCR (Roche). The sequencing was performed on a NextSeq500 device (Illumina) using either a High Output v2 Kit (150 cycles) with the recommended sequencing conditions (read1: 26nt, read2: 98nt, index1: 8 nt, index2: n.a.) or a Mid Output v2 Kit (300 cycles for BCR repertoire analysis (read1: 150 nt, read2: 150 nt, index1: 8 nt, index2: n.a., 20% PhiX spike-in).

**Single cell RNA and B cell receptor repertoire analysis.** Raw sequencing data for single cell transcriptome analysis were processed using cellranger-2.1.1. mkfastq and count commands with default parameter settings, the genome reference: refdata-cellranger-mm10-1.2.0 and expected-cells = 3000[56]. Raw sequencing data for single cell immune profiling (B cell receptor sequences (BCR) for IgG heavy and light chain) were processed using cellranger-3.0.2. mkfastq and vdj commands with default parameter settings and the vdj-reference: refdata-cellranger-vdj_GRCm38_alts_ensembl-mouse-2.2.0. Only cells, which appeared in both, the single cell transcriptome as well as immune receptor profiling were further analyzed. The high-confidence contig sequences of the isotype-switched memory B cells were reanalyzed using HighV-QUEST at IMGT web portal for immunoglobulin (IMGT) to retrieve the V-genes, J-genes, D-genes as nucleotide and amino acid CDR3 sequence[57]. IMGT-gapped-nt-sequences, V-REGION-mutation-and-AA-change-table as well as nt-mutation-statistics were used to determine the corresponding gapped germline FR1, CDR1, FR2, CDR2, FR3 sequences as well as estimated mutation counts in the FR1-FR3 region. The most abundant contig for the heavy and light BCR chain was assigned to the corresponding cell in the single cell transcriptome analysis. Cells with incomplete heavy and light chain annotation were removed from further analysis. In order to simulate random distribution and determine the significance of observed BCR distribution within the transcriptomic clusters of Bsm in the spleen and BM of the individual Bsm, their BCR were randomly reshuffled 1000 times. For the mutation rates, the sum of the estimated mutation counts in the light and heavy BCR chain in FR1-FR3 regions was normalized by the length of the corresponding FR1-FR3 sequences. The overlap of the BCR-repertoire was computed using the V and J-gene configuration and the nucleotide sequence of the CDR3-Region for assessing similarity of the BCR and the minimal fraction of common BCR in one of the compared clusters. The analysis was repeated for 1000 random annotations of the BCRs. Overlap was defined as significantly higher or lower than expected if occurred in less than 5 or more than 95% of randomizations. Clustering was performed using Ward2 and the Manhattan distance. For the BCR trajectory analysis, clonal families were clustered by the identical isotype annotation, VDJ-gene usage, gapped germline FR1-FR3 sequence and nucleotide CDR3 sequence length of the heavy and light chain. For each clonal family a lineage tree was computed using GLaMST with concatenated FR1-FR3 sequences of the heavy and light chain and the germline sequence as root as input[58]. An accumulation of mutations was assumed between a FR1–FR3 sequences and the first "ancestor" sequence found in the sample. For each cluster, the proportion of ancestor sequences (member of a clonal family with descendants with higher mutations rates) was computed. A proportion was defined as significant if it occurred in less than 50 of 1000 randomizations ($P < 0.05$). For graphical projections of trajectories, clusters were connected if a descendant-ancestor sequence relationship was found between these clusters. Significant connections between two clusters were assumed if they occured in less than 10 of the 1000 randomizations. The single cell transcriptome data was further analyzed using

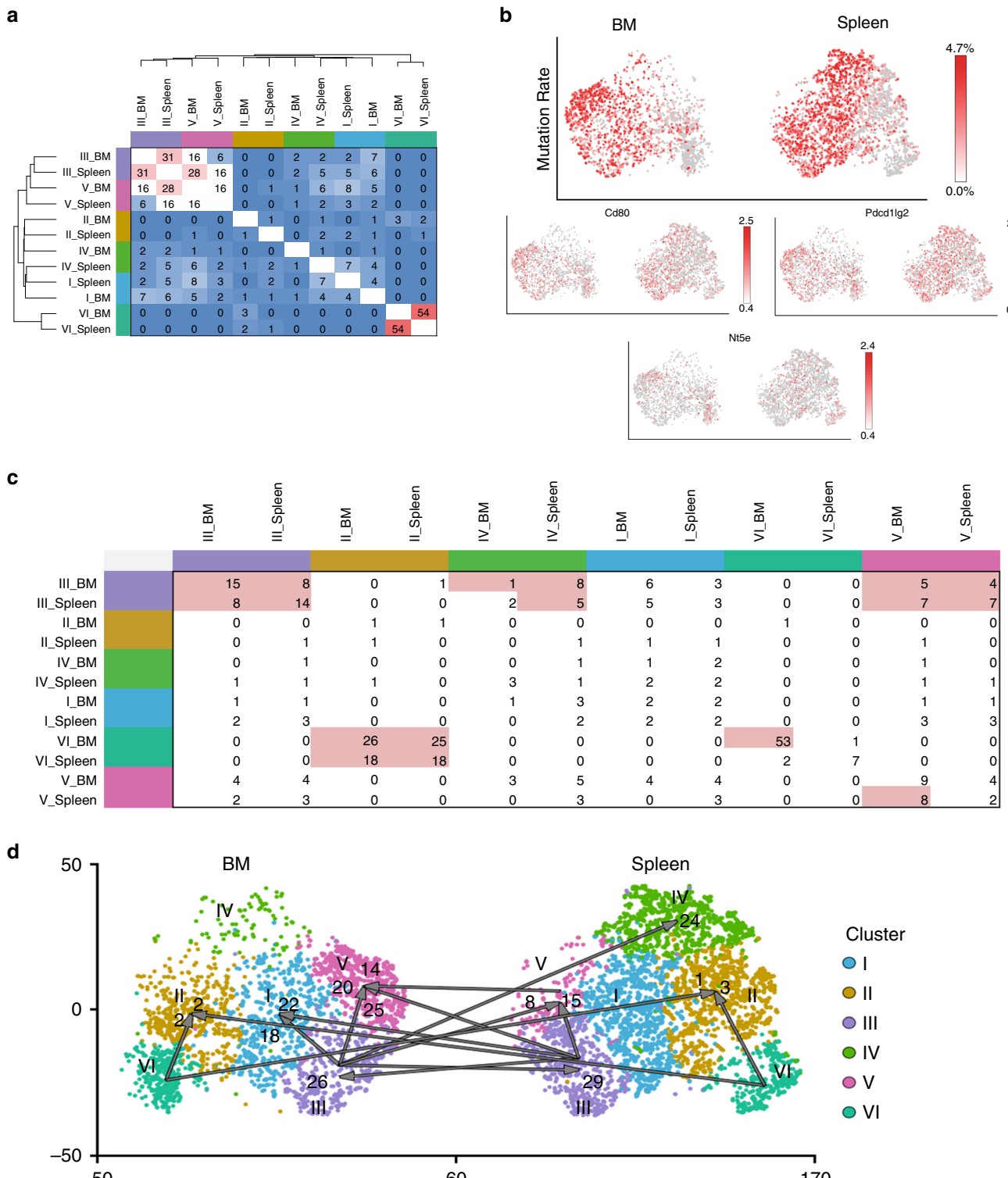

**Fig. 5 Clonal trajectories connect predominantly clusters III and V of spleen and bone marrow. a** Percentage of shared clones between clusters of spleen and bone marrow. Clones were defined according to IgG heavy and light chain genes. Clones were randomly redistributed to clusters. Overlap significantly higher than random ($P < 0.05$, randomization test) indicated by red shading, white represents expected random overlap, and blue shading overlap lower than random, i.e., exclusive repertoires. **b** Mutation rates (percent of nucleotide sequence; upper panel) and CD80 (*Cd80*), CD73 (*Nt5e*), and PD-Ligand 2 (*Pdcd1lg2*) gene expression of individual cells. Cells for single cell sequencing were isolated by FACS as IgG+CD19+CD38+CD138−GL7− small lymphocytes. **c** Frequencies of Bsm clones represented with higher mutation rates in clusters listed in rows, as compared to clusters listed in columns. Significant presence of clone members with additional mutations as compared to random distribution ($P < 0.01$, randomization test) is shaded in red. **d** Trajectories are based on accumulation of mutations within clones. Represented are significant trajectories ($P < 0.01$, randomization test). Arabic numerals indicate numbers of clones with additional mutations. Randomized redistribution of clones was performed 1000 times (**a**, **c**, **d**). Source data is provided as a Source Data file.

R 3.5.0., and Seurat R package 2.3.4. In particular, gene expression levels were normalized and the detection of variable genes, tSNE[59] as well as clustering by shared nearest neighbor modularity optimization was performed by using 5 out of 30 principle components for tSNE and a modularity of 0.6 for clustering. In addition, UMAP was performed in default parameter setting using 30 principle components. The ROC for each variable gene was computed for each cluster against the remaining cells. The ROC-values above 0.7 were clustered using Ward2 linkage, Euclidian distance. Hill numbers as well as alpha diversity measures were computed using own scripts. For sample coverage (sample completeness, SC) the iNEXT R package was used in default parameter settings using abundances of the clonotypes[60,61]. Single cell transcriptome data as well as immune profiling data data discussed in this publication are available at gene expression omnibus (GEO) under accession number GSE140133.

**Ig sequencing and repertoire analysis**. Cells from BM (tibiae, femurae and pelvic bones) and spleen of three times NP-CGG immunized-mice were isolated and Bsm were magnetically enriched using the Memory B cell Isolation Kit (130-095-838 Miltenyi). Bsm were further enriched by FACSorting of CD19+CD38+CD138− CD11c−GL7−IgM−IgD− small lymphocytes. Sorted cells of Mouse 1 and 2 were split into two equal aliquots (samples A, B) for BM and spleen samples as cellular (biological) replicates[62]. Sorted cells of Mouse 3 were split into four samples (A, B, C, D). Biological replicates were processed independently from this point on. Cell samples were lysed in RNA lysis buffer (R1060-1-50, Zymo Research) and stored at −80 °C. Total RNA was extracted from samples using the ZR RNA Miniprep Kit (Zymo Research) according to the manufacturer's protocol (Catalog nos. R1064 & R1065). Isolated RNA was split and library preparation was performed in technical duplicates.

First-strand cDNA was synthesized with SMARTScribe Reverse Transcriptase (Clontech) using total RNA, a cDNA synthesis primer mix (mIgG12ab_r1 (KKACAGTCACTGAGCTGCT), mIgG3_r (GTACAGTCACCAAGCTGCT), mIgA_r (CCAGGTCACATTCATCGTG) by Metabion international AG) and a 5′ —template-switch adapter with unique molecular identifiers (UMI) (SmartNNNa (AAGCAGUGGTAUCAACGCAGAGUNNNNUNNNNUNNNNNUCTT(rG)4)) according to the protocol "high-quality full length immunoglobulin profiling with unique molecular barcoding" by the Chudakov lab[63]. cDNA was purified with MinElute PCR purification Kit (Qiagen) and eluted in 10 μl 70 °C nuclease-free H₂O (Qiagen).

The first PCR was performed according to the protocol of the Chudakov lab[63] using a Step-out primer M1SS (AAGCAGTGGTATCAACGCA) (annealing on the switch adapter) and a mouse IgH reverse primer mix (mIgG12_r2 (ATTGGGCAGC CCTGATTAGTGGATAGACMGATG), mIgG3_r2 (ATTGGGCAGCCCTGATTAAGGGATAGA CAGATG), mIgA_r2 (ATTGGGCAGCCCTGATTTCAGTGGGTAGATGGTG) binding to the constant region of certain Ig heavy chains). The first PCR was performed with Q5 Hot Start High-Fidelity DNA polymerase (NEB) in a 50 μl reaction volume using 7.5 μl of cDNA product with following PCR parameters: 1 cycle of 95 °C for 1 min 30 s, 20 cycles of 95 °C for 10 s, 60 °C for 20 s, 72 °C for 40 s, 1 cycle of 72 °C for 4 min; storage at 4 °C. PCR 1 products were purified with MinElute PCR purification Kit (Qiagen) and eluted in 25 μl 70 °C nuclease-free H₂O (Qiagen).

Within the second PCR amplification[63] a M1S primer ((N)4–6(XXXXX) CAGTGGTATCAACGCAGAG) (annealing on M1SS) and a step-out primer Z ((N)4-6(XXXXX)ATTGGGCAGCCCTGATT), both with sample barcodes, were used. PCR 2 was performed with Q5 Hot Start High-Fidelity DNA polymerase (NEB) in a 50 μl reaction volume using 2 μl of PCR 1 product with following PCR parameters: 1 cycle of 95 °C for 1 min 30 s; 14–15 cycles of 95 °C for 10 s, 60 °C for 20 s, 72 °C for 40 s; 1 cycle of 72 °C for 4 min; storage at 4 °C. PCR 2 products were purified with MinElute PCR purification Kit (Qiagen) and eluted in 25 μl 70 °C nuclease-free H₂O (Qiagen). The products were also gel-purified from 2% agarose gels (extraction with MinElute gel extraction Kit (Qiagen); elution in 15 μl 70 °C nuclease-free H₂O (Qiagen).

Adapter ligation was performed using the TruSeq® DNA PCR-Free library Prep protocol (Illumina). The products were gel-purified from 2% agarose gels instead of bead purification as mentioned in the protocol (extraction with MinElute gel extraction Kit (Qiagen); elution in 10 μl 70 °C nuclease-free H2O (Qiagen)).

The quality of amplified libraries was verified using an Agilent 2100 Bioanalyzer (2100 expert High Sensitivity DNA Assay). According to the fragment size, the libraries were quantified by qPCR using the KAPA library Quantification Kit for Illumina platforms (KAPA Biosystems). Based on the result of the qPCR a final library pool with a concentration of 2 pM was used for sequencing with NextSeq500/550 (Illumina) using the NextSeq® 500/550 Mid Output Kit v2 (150 cycles) for 2 × 150 bp paired-end sequencing with 20% PhiX-library spike-in.

Ig repertoire analysis was performed using MIGEC-1.2.4a[64] in default parameter settings while adding a demultiplexing step for identification of IgG1/2, IgG₃ and IgA heavy chains. After the MIGEC pipeline's "checkout" step isotypes were classified according to the presence of mIgG12_r2, mIgG3_r2 and mIgA_r2 primer sequences: AGTGGATAGACMGATG, AAGGGATAGACAGATG and TCAGTGGGTAGATGGTG, allowing for one mismatch against the primer sequence. Data were then processed independently for each isotype. The MIGEC segments file was adjusted to include only C57BL/6-specific V genes for mapping. MIGEC performs a UMI-guided correction to remove PCR as well as sequencing

bias and errors. Each resulting consensus sequence was treated as one clone. Clones with identical V and J gene compositions and CDR3 nucleotide sequences were defined as clonotypes. Solely clonotypes consistently found in both technical replicates of a given sample were considered in downstream analyses. Statistics on the overlap of repertoires between different samples were performed based on the presence of clonotypes. To assess likelihood of the observed presence of clonotypes exclusively in one organ or an overlapping presence in both being the result of differences in clone numbers or rare clones at purely random distribution, clones were randomly distributed 1000 times among the samples. Random distributions of sequences to paired samples were drawn while retaining the initial samples' numbers of clones. Overlap statistics represent the median overlap and clonotypes exclusive to one sample from the 1000 randomizations. The degree of similarity between samples accounting for the abundance of clonotypes is represented by the cosine similarity[65]. Hill numbers as well as alpha diversity indices were computed using own scripts. For Sample Coverage (Sample Completeness, SC) the iNEXT R package was used in default parameter settings using abundances of pooled biological replicates for all colonotypes as well as colonotypes found (overlapping clones) and not found in the different organ (unique clones)[60,61]. Ig repertoire data discussed in this publication are available at GEO under accession number GSE140133.

**Statistics and data representation**. Absolute numbers of mouse B cell subsets per organ were calculated based on their frequency in a sample. For spleen, peripheral and mesenteric lymph nodes and Peyer's patches, total organs were prepared and the total numbers of B cell populations calculated based on the numbers in a defined volume determined by flow cytometry (MACSQuant, Miltenyi). BM cell numbers were calculated analogously based on cell numbers in a single femur of a mouse, which is estimated to harbor 6.3% of total BM leading to a conversion factor of 7.9 from two femurs to total murine BM[34].

Further analyses and statistical tests were performed within the R programming environment[66], with use of the non-base package VennDiagram[67]. Whiskers in Tukey boxplots span 1.5IQR.

**Preparation of histological sections and confocal microscopy**. Spleens and femoral bones were fixed in 4% Paraformaldehyde (Electron Microscopy Sciences) for 4 h at 4 °C, equilibrated in 30% sucrose/PBS, then frozen and stored at −80 °C. Six micrometer cryosections were stained for 1 h in 0.1% Tween-20 (Sigma-Aldrich)/10% FCS/PBS after blocking with 10% FCS/PBS for 1 h. The following primary and secondary reagents were used: anti-mouse IgG2b-AlexaFluor546 (RMG2b-1, BioLegend Cat No 406702, 1:100); anti-GFP-AlexaFluor488 (rabbit polyclonal, Life Technologies Cat No A-21311, 1:100; rat monoclonal FM264G, BioLegend Cat No 338008,1:100), anti-fibronectin (rabbit polyclonal, Sigma Aldrich Cat No F3648,1:100), biotinylated anti-mouse Ki67 (Sol-15, eBioscience Cat No 13-5698-82, 1:100), biotinylated anti-mouse VCAM-1 (429, eBioScience Cat No 13-1061-82, 1:100), anti-human/mouse cadherin 17 (rabbit polyclonal, R&D Cat No AF8524, 1:50), anti-mouse laminin (rabbit polyclonal, Sigma Aldrich L9393, 1:100), anti-mouse IgD-AlexaFluor647 (11.26c, DRFZ), anti-mouse Thy1-Alexa Fluor 594 (T24, DRFZ), anti-mouse B220-AlexaFluor594/647 (RA3.6B4, DRFZ), anti-mouse CD11c-AlexaFluor647 (N418, DRFZ), donkey anti-rabbit polyclonal IgG-AlexaFluor488/647 (Thermo Fisher Cat No A-21206,1:600), strepatavidin-AlexaFluor594/647 (Thermo Fisher S323561:1000). For nuclear staining, sections were stained with 1 μg per ml DAPI in PBS. Sections were mounted in Fluorescent Mounting Medium (DAKO). For confocal microscopy, a Zeiss LSM710 with a 20×/0.8 numerical aperture objective lens was used. Images were generated by tile-scans and maximum intensity projection of 3–5 Z-stacks each with 1 μm thickness. Image acquisition was performed using Zen 2010 Version 6.0 and images were analyzed by Zen 2012 Light Edition software (Carl Zeiss MicroImaging). Co-localization of Bsm with cells expressing other markers was performed using sections from three biological replicates. More than 40 cells were inspected per analysis.

**Manual image analysis**. Nearest neighbors of Bsm were enumerated based on either direct cell-cell contact or a position within a 10 μm radius of cell boundaries of Bsm using high resolution images acquired by confocal microscopy. Cells of interest were identified by immunofluorescent staining for the marker in question. Cells were counted as neighboring cells if pixels from both cells were in either direct contact or within the 10 μm radius of cell boundaries.

**Modeling random co-localization**. To determine the probability of Bsm cells randomly co-localizing to their observed nearest neighbors, we employed modeling of random cell positioning, in a modification of the previously published approach[8]. In brief, images of Bsm were positioned on histological images of BM at random, repeatedly. Frequencies of co-localizing Bsm and stromal cells were then determined and compared to the frequencies of the original histological images.

**Reporting summary**. Further information on research design is available in the Nature Research Reporting Summary linked to this article.

## Data availability

Next Generation Sequencing data sets generated and analyzed during the current study are available in the GEO repository under accession number GSE140133. All other data that support the findings of this study are available on request from the corresponding author [M.F.M.].

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

## Acknowledgements

We thank Tuula Geske, Heidi Hecker-Kia, and Heidi Schliemann for expert technical help, Toralf Kaiser and Jenny Kirsch for support in cell sorting, and Manuela Ohde, Vivien Theißig and Patrick Thiemann for animal care. We thank Sven Künzel and Christine Pfeifle, MPI for Evolutionary Biology in Plön, for providing help and expertise on wild mice. We would like to thank Dr. Dmitriy M. Chudakov for kindly providing the protocol for the generation of libraries for BCR receptor repertoire sequencing. This work was supported by Deutsche Forschungsgemeinschaft through DFG priority program 1468 IMMUNOBONE (to A.R. and H.D.C.) and the TR130 (to A.R. and H.D.C.), and by the European Research Council through the Advanced Grant IMMEMO (ERC-2010-AdG.20100317 Grant 268978 to A.R.). A.E.H. was funded by the DFG (TRR130, HA5354/6-1 and HA5354/8-1). The work of R.R. and D.S. was funded in part by the Berlin-Brandenburg School for Regenerative Therapies. R.A. was a member of Berlin-Brandenburg School for Regenerative Therapies. J.Z. was a member of GRK1121 (ZIBI) and M.W., K.W., and S.D. were members of International Max Planck Research School for Infectious Diseases and Immunology Berlin (IMPRS-IDI). M.F.M. is supported by e:Bio Innovationswettbewerb Systembiologie, a program of the Federal Ministry of Education. Work was supported by the state of Berlin and the "European Regional Development Fund" to G.A.H., F.H., P.M., M.M., and M.F.M. (ERDF 2014–2020, EFRE 1.8/11, Deutsches Rheuma-Forschungszentrum). HDC is funded by the Dr. Rolf M. Schwiete Foundation. The work of V.G., U.M., and S.T.R. work was funded in part by the Swiss National Science Foundation (Project #: 31003A_143869, 31003A_170110 to STR), SystemsX.ch—AntibodyX RTD project (to S.T.R.), Swiss Vaccine Research Institute (to S.T.R.). P.M. is supported by EUTRAIN, a FP7 Marie Curie Initial Training Network for Early Stage Researchers funded by the European Union. The DRFZ is a Leibniz Institute.

## Author contributions

Conceptualization: R.R., M.F.M., A.R., H.D.C.; Methodology: R.R., P.D., M.F.M., A.R.; Software: R.R., F.H., P.D.; Validation: R.R., R.A., M.F.G., D.S., C.K., R.K., J.S.; Formal analysis: R.R., P.D., R.K., J.S., V.G., St.K., U.S.; Investigation: R.R., R.A., M.F.G., G.A.H., J.K., V.G., U.M., R.K., D.S., C.K., C.H., Si.K., K.L., P.M., M.M.G., S.N., S.H., Ö.S.A., F.S., J.S., M.W., K.W., and J.Z.; Resources: V.G., R.K., Ö.S.A., R.C., A.E.H., and S.T.R.; Data curation: R.R., F.H., P.D.; Writing—original draft: R.R., R.A., H.D.C., M.F.M., and A.R.; Writing—Review & editing: M.F.G., V.G., A.E.H., P.D., U.M., F.S.; Visualization: R.R., P.D., R.A., M.F.M.; Supervision: A.R., M.F.M., H.D.C.; Project administration: M.F.M., A.R., H.D.C.; Funding acquisition: A.R., M.F.M.

## Competing interests

The authors declare no competing interests.
