## [Peer Review File · Nature Communications]

Reviewers' comments:

Reviewer #1 (Systems immunology, transcriptome analyses)(Remarks to the Author):

The overall quality of the work being presented is very good. There are no major comments from my end. Documenting the compartmentalization, heterogeneity and fate of Bsm is an important contribution. The discovery and description of BM-specific and spleen-specific populations is of course of particular interest. We are just left wishing we could know a little more about the functional implications of those findings. Maybe this could at least be discussed. The work is overall rather descriptive and maybe more could have been done, especially in an animal model setting. Additional points otherwise include the following:

- The introduction was confusing as it was difficult to dissociate background information from question / gap that the work was trying to address (and how it was going to go about it). This could make it difficult for someone not already immersed in the B-cell literature to form an opinion on the potential importance of the work being undertaken.
- The authors intend to contribute the transcriptome profiling data to GEO, which should be applauded. It would be good though if a GSE ID and token code be provided along with the revised manuscript. This permits reviewer access and verification for instance that an adequate level of descriptive information is provided (while keeping the dataset private to all other users).
- It could be useful to discuss relevance to human immunology. To what extent would these findings apply in humans? What would be the implications?
- The read depth and overall number of cells profiled in the scRNAseq experiment is on the low end, but likely adequate for simple phenotyping of populations. Are there novel markers that could be identified and aid with the characterization of BM-specific and spleen-specific populations?

Reviewer #2 (B cell biology, BCR signalling)(Remarks to the Author):

This manuscript by Riedel and colleagues describes memory B cell heterogeneity in conventionally housed, pet shop, and feral mice. These subsets were defined by repertoire analysis, macro- and microanatomical localization, and transcriptional programs. Overall, these findings are likely to be of value to the B cell community.

The review below is organized as per the criteria and format proposed by Krummel et al (Universal Principled Review: A Community-Driven Method to Improve Peer Review, Cell, 2019).

Objective Criteria (Quality)

1 Quality: Experiments (1–3 scale): 2

- Figure by figure, do experiments, as performed, have the proper controls?

-Yes.

- Are specific analyses performed using methods that are consistent with answering the specific question?

-The study is inherently discovery-based and descriptive. As such, the methods and analyses used are appropriate to define memory B cell heterogeneity.

•Is there the appropriate technical expertise in the collection and analysis of data presented?

-Yes

•Do analyses use the best-possible (most unambiguous) available methods quantified via appropriate statistical comparisons?

-One newer and more reproducible method for analyzing and displaying scRNA-seq data is UMAP. This method is objectively superior to tSNE analysis in that the distances between clusters are quantitatively meaningful (Becht et al, Nat Biotech, 2019).

-The claim of 'resident' memory B cells in the spleen vs. bone marrow needs additional statistical analysis and evidence of sufficient sampling. The authors' claim that biological splenic memory B cell replicates are more similar than within-animal comparisons between spleen and bone marrow subsets is reasonable. That said, an estimation of the diversity of these subsets is needed (e.g. ecological diversity indices). If a subset of memory cells is exceptionally diverse, then finding CDR3 overlap is intrinsically unlikely, irrespective of the underlying biology of residency vs. circulation. In these cases, efforts must be reported to ensure proper sampling of the repertoire. The number of unique clonotypes per cells sorted should also be reported.

•Are controls or experimental foundations consistent with established findings in the field? A review that raises concerns regarding inconsistency with widely reproduced observations should list at least two examples in the literature of such results. Addressing this question may occasionally require a supplemental figure that, for example, re-graphs multi-axis data from the primary figure using established axes or gating strategies to demonstrate how results in this paper line up with established understandings. It should not be necessary to defend exactly why these may be different from established truths, although doing so may increase the impact of the study and discussion of discrepancies is an important aspect of scholarship.

-Yes

2. Quality: Completeness (1–3 scale): 2

•Does the collection of experiments and associated analysis of data support the proposed title- and abstract-level conclusions? Typically, the major (title- or abstract-level) conclusions are expected to be supported by at least two experimental systems.

-In general yes (though see comments below). Heterogeneity is defined through unbiased transcriptional analyses and IgH CDR3 repertoires. Confirmation by flow cytometry of some more of the scRNA-seq differences (e.g. CD55, Itga4) would be helpful.

•Are there experiments or analyses that have not been performed but if "true" would disprove the conclusion (sometimes considered a fatal flaw in the study)? In some cases, a reviewer may propose an alternative conclusion and abstract that is clearly defensible with the experiments as presented, and one solution to "completeness" here should always be to temper an abstract or remove a conclusion and to discuss this alternative in the discussion section.

-Yes. While the demonstration of memory B cell heterogeneity is persuasive, an underlying theme throughout the paper and title is that there are distinct resident memory B cells that are maintained independently of one another in the spleen and bone marrow. But there is no direct experimental proof of this--only data that are suggestive based on repertoire analyses (some of which needs to be

reanalyzed for diversity and sufficient sampling as per my comments above). Previous studies on resident vs. circulating memory lymphocytes, such as that reported by Randall and colleagues (Allie et al, Nature Immunology, 2019), used direct experimental methods to confirm such a claim, including parabiosis, perfusions, adoptive transfers, etc. This study does not meet this bar established by previous Trm and B cell memory studies.

3. Quality: Reproducibility (1–3 scale): 1

- Figure by figure, were experiments repeated per a standard of 3× repeats or 5 mice per cohort, etc.?

-Yes, when feasible. scRNA-seq experiments are far too expensive to be repeated in such ways.

- Is there sufficient raw data presented to assess rigor of the analysis?

-Yes, example flow cytometry plots are shown as are representative images. A precondition of eventual publication should be deposition of scRNA-seq data on a publicly available database.

- Are methods for experimentation and analysis adequately outlined to permit reproducibility?

-Yes.

4. Quality: Scholarship (1–4 scale but generally not the basis for acceptance or rejection): 3

- Has the author cited and discussed the merits of the relevant data that would argue against their conclusion?

-In part. The major paper with which this work disagrees is that of Giesecke et al. (JI, 2014)—a human study that found little heterogeneity in MBCs across different organs. This earlier study used a fairly limited set of markers and criteria to make their conclusions. The present study, although performed in mice and not strictly comparable, uses modern unbiased approaches to reach different conclusions.

-Some discussion is needed of work by Wang and McHeyzer-Williams, which reported differences in Rora and Tbet in memory B cells of different isotypes.

- Has the author cited and/or discussed the important works that are consistent with their conclusion and that a reader should be especially familiar when considering the work?

-Discussion of Troy Randall's recent study on resident B cell memory is a notable omission.

-Prior studies on memory B cell heterogeneity at the marker and functional level should also be cited and discussed (e.g. Zuccarino-Catania, NI, 2014; Dogan et al., NI, 2009; Krishnamurthy et al., Immunity, 2016; Pape et al. Science, 2011; Seifert et al., PNAS, 2015).

- Specific (helpful) comments on grammar, diction, paper structure, or data presentation (e.g., change a graph style or color scheme) go in this section, but scores in this area should not to be significant bases for decisions.

-The discussion of the scRNA-seq data (pages 9-10) should be made more concise.

More Subjective Criteria (Impact)

5. Impact: Novelty/Fundamental and Broad Interest (1–4 scale): 3

- A score here should be accompanied by a statement delineating the most interesting and/or important conceptual finding(s), as they stand right now with the current scope of the paper. A “1” would be expected to be understood for the importance by a layperson but would also be of top interest (have lasting impact) on the field.

-The unbiased scRNA-seq-based identification of memory B cell heterogeneity will be useful as a resource to the community.

-The impact would of course be greatly strengthened by functional studies to help explain what this transcriptional heterogeneity means--how do these newly identified subsets differentially behave upon re-challenge? How do they compare with previously established heterogeneity by other groups and conventional markers? That said, such studies would likely exceed what would typically be reported in a Nature Communications article.

- How big of an advance would you consider the findings to be if fully supported but not extended? It would be appropriate to cite literature to provide context for evaluating the advance. However, great care must be taken to avoid exaggerating what is known comparing these findings to the current dogma (see Box 2). Citations (figure by figure) are essential here.

-To my knowledge the findings are new. I am aware of only one other study by Quake and colleagues (Science, 2018) that applied scRNA-seq to memory B cells. This human study focused on IgE and bears little relevance to the current study.

Point-by-point response to the reviewers' comments:

Reviewer #1 (Systems immunology, transcriptome analyses) (Remarks to the Author):

The overall quality of the work being presented is very good. There are no major comments from my end. Documenting the compartmentalization, heterogeneity and fate of Bsm is an important contribution. The discovery and description of BM-specific and spleen-specific populations is of course of particular interest. We are just left wishing we could know a little more about the functional implications of those findings. Maybe this could at least be discussed. The work is overall rather descriptive and maybe more could have been done, especially in an animal model setting.

Response: We thank the reviewer for the compliments.

*We agree that the observed heterogeneity and residency of B cell memory opens a new field for research asking for functional implications. Comprehensive studies, in particular in vivo studies, are currently underway in our group. This extensive line of experiments would justify a separate manuscript and is beyond the scope of the present manuscript. We consider it important to communicate our findings now to the community, anticipating that this will stimulate research of other groups. Indeed, we are **providing already now a stimulating original hypothesis**, suggesting that bone marrow resident Bsm may be direct precursors of bone marrow plasma cells (page 17, line 349-357).*

Additional points otherwise include the following:

- The introduction was confusing as it was difficult to dissociate background information from question / gap that the work was trying to address (and how it was going to go about it). This could make it difficult for someone not already immersed in the B-cell literature to form an opinion on the potential importance of the work being undertaken.

*Response: **We have edited the introduction** to provide some background and clearly state the fundamental question addressed (revised manuscript p.4-5).*

- The authors intend to contribute the transcriptome profiling data to GEO, which should be applauded. It would be good though if a GSE ID and token code be provided along with the revised manuscript. This permits reviewer access and verification for instance that an adequate level of descriptive information is provided (while keeping the dataset private to all other users).

*Response: The sequencing data have been deposited to GEO – GSE140133 with token (kpihqekjhwvpcd). GEO IDs **are now included** into the Materials and Methods section (page 23, line 469; page 26, line 544).*

- It could be useful to discuss relevance to human immunology. To what extent would these findings apply in humans? What would be the implications?

Response: In principle, the organization of immunological memory in humans and mice is fairly similar, as we and others have shown for memory plasma cells and memory T lymphocytes (Tokoyoda et al., Immunity 2009, Sercan-Alp et al., EJI 2015, Arce et al., J Leukoc

*Biol, 2004, reviewed in Chang et al., Immunol Rev, 2018). For memory T lymphocytes, we had shown before (Okhrimenko et al., PNAS 2014), and this could only be done for humans, that they preferentially conserve long-term memory to systemic challenges, e.g. measles, mumps, rubella. This would be our hypothesis also for Bsm, but presently no data are available on this. We have **mentioned this in the discussion** now (page 17, line 332-335).*

- The read depth and overall number of cells profiled in the scRNAseq experiment is on the low end, but likely adequate for simple phenotyping of populations. Are there novel markers that could be identified and aid with the characterization of BM-specific and spleen-specific populations?

*Response: In Figure 3d and 3e, we present several genes both encoding for surface and intracellular proteins, which can either be used as markers alone or in combination in order to identify and characterize the BM or the spleen specific Bsm populations. In **new Figure 4** we validate several of these markers (CD5, CD49d, CXCR3 and CD11b) on the protein level by immunofluorescence, thus allowing unambiguous cytometric identification and isolation of Bsm of each of the six clusters.*

Reviewer #2 (B cell biology, BCR signalling)(Remarks to the Author):

This manuscript by Riedel and colleagues describes memory B cell heterogeneity in conventionally housed, pet shop, and feral mice. These subsets were defined by repertoire analysis, macro- and microanatomical localization, and transcriptional programs. Overall, these findings are likely to be of value to the B cell community.

The review below is organized as per the criteria and format proposed by Krummel et al (Universal Principled Review: A Community-Driven Method to Improve Peer Review, Cell, 2019).

Objective Criteria (Quality)

1 Quality: Experiments (1–3 scale): 2

- Figure by figure, do experiments, as performed, have the proper controls?

-Yes.

- Are specific analyses performed using methods that are consistent with answering the specific question?

-The study is inherently discovery-based and descriptive. As such, the methods and analyses used are appropriate to define memory B cell heterogeneity.

- Is there the appropriate technical expertise in the collection and analysis of data presented?

-Yes

- Do analyses use the best-possible (most unambiguous) available methods quantified via appropriate statistical comparisons?

-One newer and more reproducible method for analyzing and displaying scRNA-seq data is UMAP. This method is objectively superior to tSNE analysis in that the distances between clusters are quantitatively meaningful (Becht et al, Nat Biotech, 2019).

*Response: We agree with the reviewer, that UMAP is useful to emphasize the distances between clusters. We have used tSNE in the first place, because it probably better reflects differences in the expression of individual cells and genes (van der Maaten, et al., Journal of Machine Learning Research, 2008, Kobak et al., bioRxiv, 2019). Nevertheless, we have now also mapped the clustering to **UMAP (new Supplemental Figure 5)**. This results in a separation of clusters comparable to tSNE. With respect to trajectories, those are based in the present manuscript on somatic hypermutation of antigen receptors, an opportunity that Bsm provide exclusively, rather than the transcriptional profiles used for clustering and dimensionality reduction.*

-The claim of 'resident' memory B cells in the spleen vs. bone marrow needs additional statistical analysis and evidence of sufficient sampling. The authors' claim that biological splenic memory B cell replicates are more similar than within-animal comparisons between spleen and bone marrow subsets is reasonable. That said, an estimation of the diversity of these subsets is needed (e.g. ecological diversity indices). If a subset of memory cells is exceptionally diverse, then finding CDR3 overlap is intrinsically unlikely, irrespective of the underlying biology of residency vs. circulation. In these cases, efforts must be reported to ensure proper sampling of the repertoire. The number of unique clonotypes per cells sorted should also be reported.

*Response: We have now added the **diversity vs hill order plots (Supplementary Figure 2c)**. Common **ecological alpha diversity indices and raw counts** are provided in **supplementary table 1**.*

*We had shown that the organ-specific repertoires do not just consist of rare clonotypes by comparing observed-to-random distributions (supplementary Figure 2 a-d). From the newly provided diversity analyses it now is evident that Bsm of bone marrow and spleen have comparable clonotype diversity, with the **samples representing than 97%-98% of the estimated entire repertoire each (supplementary Figure 2c and 3)**. Clonotypes found in both organs represent more of the estimated full repertoires (98.3% - 99.9%) than organ specific clonotypes (74%-95.9%). Furthermore, the scatterplot of clonal frequencies shows that the abundance of clonotypes found only in spleen or bone marrow or in both organs is comparable (**new supplementary Figure 2d**), indicating that the observed overlap is not biased by sampling. The numbers of unique clonotypes are reported in **new supplementary Table 1**.*

•Are controls or experimental foundations consistent with established findings in the field?
A review that raises concerns regarding inconsistency with widely reproduced observations should list at least two examples in the literature of such results. Addressing this question may occasionally require a supplemental figure that, for example, re-graphs multi-axis data from the primary figure using established axes or gating strategies to demonstrate how results in this paper line up with established understandings. It should not be necessary to defend exactly why these may be different from established truths, although doing so may increase the impact of the study and discussion of discrepancies is an important aspect of scholarship.

-Yes

2. Quality: Completeness (1–3 scale): 2

•Does the collection of experiments and associated analysis of data support the proposed title- and abstract-level conclusions? Typically, the major (title- or abstract-level) conclusions are expected to be supported by at least two experimental systems.

-In general yes (though see comments below). Heterogeneity is defined through unbiased transcriptional analyses and IgH CDR3 repertoires. Confirmation by flow cytometry of some more of the scRNA-seq differences (e.g. CD55, Itga4) would be helpful.

Response, see also response to reviewer #1: In addition to the IgH CDR3 repertoire analyses and the scRNA transcriptome analyses we now use signature genes expressed on the cell surface to confirm by immunofluorescence and flow cytometry the heterogeneity observed by scRNA-seq (new Figure 4). All of the identified scRNA-seq clusters can be identified, and eventually isolated, according to expression of cell surface markers (gated on Bsm) CD21/35, CD23, CD5, and CD49d. We consider this an important extension of our previously shown data.

•Are there experiments or analyses that have not been performed but if “true” would disprove the conclusion (sometimes considered a fatal flaw in the study)? In some cases, a reviewer may propose an alternative conclusion and abstract that is clearly defensible with the experiments as presented, and one solution to “completeness” here should always be to temper an abstract or remove a conclusion and to discuss this alternative in the discussion section.

-Yes. While the demonstration of memory B cell heterogeneity is persuasive, an underlying theme throughout the paper and title is that there are distinct resident memory B cells that are maintained independently of one another in the spleen and bone marrow. But there is no direct experimental proof of this--only data that are suggestive based on repertoire analyses (some of which needs to be reanalyzed for diversity and sufficient sampling as per my comments above). Previous studies on resident vs. circulating memory lymphocytes, such as that reported by Randall and colleagues (Allie et al, Nature Immunology, 2019), used direct experimental methods to confirm such a claim, including parabiosis, perfusions, adoptive transfers, etc. This study does not meet this bar established by previous Trm and B cell memory studies.

*Response: We argue that actually **different repertoires of antigen receptors are the best available evidence documenting different compartments of memory lymphocytes**. The diversity of antigen-receptors and clonal selection in immune reactions shape the repertoire of memory lymphocytes. If those were one circulating population, they essentially must have one repertoire. If they were maintained in different compartments, they could but not necessarily must have different repertoires. Different repertoires, and this is what we find, clearly and unambiguously show (Szabo et al, Science Immunology, 2019) that the clusters identified represent different compartments!*

It should be noted on the side that parabiosis experiments cannot be performed in Germany. With respect to perfusions, bone marrow and spleen are immediately penetrated by dyes and antibody-conjugates, thus not allowing to identify resident cells this way (own results, unpublished). Neither can adoptive transfer, since tissue-resident memory cells, when dislocated and transferred into the blood, apparently can home to a variety of tissues (Tokoyoda et al., Immunity, 2009).

We sincerely hope that the reviewer can agree on our line of argumentation.

*The work of Allie et al. we have **now cited**.*

3. Quality: Reproducibility (1–3 scale): 1

- Figure by figure, were experiments repeated per a standard of 3× repeats or 5 mice per cohort, etc.?

-Yes, when feasible. scRNA-seq experiments are far too expensive to be repeated in such ways.

- Is there sufficient raw data presented to assess rigor of the analysis?

-Yes, example flow cytometry plots are shown as are representative images. A precondition of eventual publication should be deposition of scRNA-seq data on a publicly available database.

Response: Done, see response to reviewer #1: (GEO – GSE140133 with token (kpihqekjhwvpcd)).

- Are methods for experimentation and analysis adequately outlined to permit reproducibility?

-Yes.

4. Quality: Scholarship (1–4 scale but generally not the basis for acceptance or rejection): 3

- Has the author cited and discussed the merits of the relevant data that would argue against their conclusion?

-In part. The major paper with which this work disagrees is that of Giesecke et al. (JI, 2014)—a human study that found little heterogeneity in MBCs across different organs. This earlier study used a fairly limited set of markers and criteria to make their conclusions. The present study, although performed in mice and not strictly comparable, uses modern unbiased approaches to reach different conclusions.

-Some discussion is needed of work by Wang and McHeyzer-Williams, which reported differences in Rora and Tbet in memory B cells of different isotypes.

*Response: We have **cited and discuss** the work of Wang and McHeyzer Williams (citation number 42, page 15, line 294).*

- Has the author cited and/or discussed the important works that are consistent with their conclusion and that a reader should be especially familiar when considering the work?

-Discussion of Troy Randall's recent study on resident B cell memory is a notable omission.

-Prior studies on memory B cell heterogeneity at the marker and functional level should also be cited and discussed (e.g. Zuccarino-Catania, NI, 2014; Dogan et al., NI, 2009; Krishnamurthy et al., Immunity, 2016; Pape et al. Science, 2011; Seifert et al., PNAS, 2015).

*Response: We have **now cited Randall's** recent seminal study demonstrating tissue-resident Bsm of the lung, while here we originally demonstrate tissue-resident Bsm of the bone marrow and the spleen, and decipher their heterogeneity on the single cell level.*

*We also briefly **mention now the work of Dogan et al., NI, 2009, Krishnamurthy et al., Immunity, 2016; Pape et al. Science, 2011; Seifert et al., PNAS, 2015, untangling the role of IgM Bm versus Bsm in secondary immune reactions, and pointing to a dichotomy of IgM Bm contributing the secondary germinal center reactions rather than direct differentiation into plasmablasts of Bsm. The role of CD80 and PD-L2 expressing Bsm as dedicated plasmablast***

precursors (Zuccarino-Catania, NI, 2014) is now also discussed in more detail. We find CD80 and PD-L2 preferentially expressed by clusters III, IV and V, i.e. in support of our hypothesis that clusters IV and V are tissue-resident precursors of plasma cells.

- Specific (helpful) comments on grammar, diction, paper structure, or data presentation (e.g., change a graph style or color scheme) go in this section, but scores in this area should not be significant bases for decisions.
- The discussion of the scRNA-seq data (pages 9-10) should be made more concise.

Response: We now have edited this part of the results section (page 9-11, line 134-200).

More Subjective Criteria (Impact)

5. Impact: Novelty/Fundamental and Broad Interest (1–4 scale): 3

- A score here should be accompanied by a statement delineating the most interesting and/or important conceptual finding(s), as they stand right now with the current scope of the paper. A “1” would be expected to be understood for the importance by a layperson but would also be of top interest (have lasting impact) on the field.
- The unbiased scRNA-seq-based identification of memory B cell heterogeneity will be useful as a resource to the community.
- The impact would of course be greatly strengthened by functional studies to help explain what this transcriptional heterogeneity means--how do these newly identified subsets differentially behave upon re-challenge? How do they compare with previously established heterogeneity by other groups and conventional markers? That said, such studies would likely exceed what would typically be reported in a Nature Communications article.

Response: We agree with the reviewer that a comprehensive study of the functional implications of the memory B cell subsets identified here would be desirable but also is beyond the scope of the present manuscript. The current manuscript provides a significant conceptual advance in our understanding of the organization and maintenance of immunological memory and should be highly stimulating for the biomedical community.

- How big of an advance would you consider the findings to be if fully supported but not extended? It would be appropriate to cite literature to provide context for evaluating the advance. However, great care must be taken to avoid exaggerating what is known comparing these findings to the current dogma (see Box 2). Citations (figure by figure) are essential here.
- To my knowledge the findings are new. I am aware of only one other study by Quake and colleagues (Science, 2018) that applied scRNA-seq to memory B cells. This human study focused on IgE and bears little relevance to the current study.

REVIEWERS' COMMENTS:

Reviewer #1 (Remarks to the Author):

GEO deposition seems to be in order. A metadata annotation spreadsheet would be helpful though.

For information:

<https://www.ncbi.nlm.nih.gov/geo/info/spreadsheet.html>

""Metadata' refers to descriptive information and protocols for the overall experiment and individual Samples. This information is supplied by completing all fields of the appropriate metadata spreadsheet template which can be downloaded from the GEOarchive templates and examples section below...."

Other points were adequately addressed. No further comments from me.

Reviewer #2 (Remarks to the Author):

The authors have addressed all my concerns.

Point-by-point response:

REVIEWERS' COMMENTS:

Reviewer #1 (Remarks to the Author):

GEO deposition seems to be in order. A metadata annotation spreadsheet would be helpful though.

For information:

<https://www.ncbi.nlm.nih.gov/geo/info/spreadsheet.html>

"Metadata' refers to descriptive information and protocols for the overall experiment and individual Samples. This information is supplied by completing all fields of the appropriate metadata spreadsheet template which can be downloaded from the GEOarchive templates and examples section below...."

Response:

The metadata file requested by the reviewer has indeed been used for upload of the data according to the requirements of GEO. All metadata contained in the spreadsheets are accessible at the GEO website with super series GSE140133 (GSE139835: "Ig repertoire of IgG+ murine memory B cells" and GSE139836: "Single-Cell transcriptomes and immune profiling of IgG+ murine memory B cells from Bone Marrow and Spleen").

Additionally, we provide the metadata spreadsheets (Supplementary Data Set) for the discretion of the reviewer.

Other points were adequately addressed. No further comments from me.

Reviewer #2 (Remarks to the Author):

The authors have addressed all my concerns.